# The Inhomogeneous Gaussian Free Field, with application to ground state correlations of trapped 1d Bose gases

**Yannis Brun[⋆] and Jérôme Dubail[†]**

CNRS & LPCT-UMR 7019, Université de Lorraine, F-54506 Vandoeuvre-lès-Nancy, France

⋆ yannis.brun@univ-lorraine.fr
† jerome.dubail@univ-lorraine.fr

## Abstract

Motivated by the calculation of correlation functions in inhomogeneous one-dimensional (1d) quantum systems, the 2d Inhomogeneous Gaussian Free Field (IGFF) is studied and solved. The IGFF is defined in a domain $\Omega \subset \mathbb{R}^2$ equipped with a conformal class of metrics $[g]$ and with a real positive coupling constant $K : \Omega \to \mathbb{R}_{>0}$ by the action $\mathcal{S}[h] = \frac{1}{8\pi} \int_\Omega \frac{\sqrt{g}d^2x}{K(x)} g^{ij}(\partial_i h)(\partial_j h)$. All correlations functions of the IGFF are expressible in terms of the Green's functions of generalized Poisson operators that are familiar from 2d electrostatics in media with spatially varying dielectric constants.

This formalism is then applied to the study of ground state correlations of the Lieb-Liniger gas trapped in an external potential $V(x)$. Relations with previous works on inhomogeneous Luttinger liquids are discussed. The main innovation here is in the identification of local observables $\hat{O}(x)$ in the microscopic model with their field theory counterparts $\partial_x h, e^{ih(x)}, e^{-ih(x)}$, etc., which involve non-universal coefficients that themselves depend on position — a fact that, to the best of our knowledge, was overlooked in previous works on correlation functions of inhomogeneous Luttinger liquids —, and that can be calculated thanks to Bethe Ansatz form factors formulae available for the homogeneous Lieb-Liniger model. Combining those position-dependent coefficients with the correlation functions of the IGFF, ground state correlation functions of the trapped gas are obtained. Numerical checks from DMRG are provided for density-density correlations and for the one-particle density matrix, showing excellent agreement.

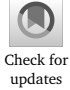

# 1 Introduction

Most gapless 1d quantum systems fall into the Luttinger liquid universality class, an effective field theory approach that accounts for their low-energy (or large distance) excitations [1–5]. This paradigm is well known for being intimately related to certain 2d conformal field theories (CFT) [6] with central charge $c = 1$, namely free massless boson theories at different compactification radii, that are themselves at the heart of the Coulomb gas picture of 2d statistical models developed in the 1970s and 1980s [7–10]. Nowadays, those free theories are playing a fundamental role in modern mathematics, especially at the intersection of probability theory and conformal geometry, where they are known as the "Gaussian Free Field" (GFF).[1] [11]

While Luttinger liquids have been studied extensively in homogeneous, translation invariant, situations, the present paper follows on from the recent series [12–20] that aims at extending the free boson CFT, or GFF, to inhomogeneous situations. [Troughout this work, inhomogeneity is understood as spatial dependence of physical quantities and parameters.] This is motivated, in part, by problems of ultracold gases in trapping potentials, see e.g. Refs. [21–32] or the discussion in Sec. 1.1 below.

So far, in the series [12–20], the focus was on those systems that possess a Luttinger parameter $K$ — a parameter that appears in the effective large-scale description and encodes the interaction strength in the 1d quantum system, see e.g. Refs. [4, 5] — that is *constant*. In that case, the inhomogeneous Luttinger liquid is nothing but a 2d CFT in a curved metric, a

---

[1]In this paper we adopt the terminology "GFF" introduced by mathematicians [11], as it has now become standard (The "GFF" has a wikipedia page: https://en.wikipedia.org/wiki/Gaussian_free_field). In physics, the GFF is known under other names such as "massless free boson", or "massless free scalar field".

fact that can be exploited to easily get nice exact analytic formulae in a variety of interesting physical situations, see Refs. [12–20].

In this paper, our goal is to explore the case where the assumption of a constant parameter $K$ is relaxed. This is natural in many physically relevant situations. Perhaps the most notable example is that of a 1d gas of bosons, modeled by the Lieb-Liniger model [33], trapped in an external potential $V(x)$, where $x$ is the spatial coordinate. In this model, the Luttinger parameter $K$ acquires a spatial dependence,

$$K \to K(x).$$

As we will explain shortly, contrary to the case of constant Luttinger parameter $K$, the underlying field theory is no longer a GFF. Instead, it is an "inhomogeneous" generalization of the GFF, with a spatially varying coupling constant, which we will dub "Inhomogeneous GFF" (IGFF). Because the IGFF is a free (or Gaussian) theory, calculating correlation functions in the IGFF boils down to solving some boundary value problem by calculating its Green's function. This will be discussed in full detail in Sec. 2. In Sec. 3, we will apply that formalism to calculate ground state correlation functions in the trapped Lieb-Liniger model.

In the rest of this introduction, we explain how exactly this work differs from previous ones on inhomogeneous Luttinger liquids, and then motivate the introduction of the IGFF, defined by the action (8) below.

## 1.1 Relation with previous works on inhomogeneous Luttinger liquids

Over the past twenty years, some of the results we will derive or use in this paper have been partially reported in the literature. Here, we give a brief account of the existing works that aimed at the same direction, to the best of our knowledge.

In 1995, Maslov and Stone [34] and (independently) Safi and Schulz [35] investigated the Landauer conductance of an interacting electron wire. Both ends of the wire are connected to a lead, represented by free electrons. In that setup, the Luttinger parameter jumps from $K = 1$ in the leads to some value fixed by the interactions in the wire. So does the velocity $v$ of gapless excitations, jumping from the Fermi velocity in the leads to some other value in the wire. Thus, the problem of calculating reflection and transmission coefficients reduces to studying the Luttinger liquid Hamiltonian with $K(x)$ and $v(x)$ that are step functions. To our knowledge, this is the first occurence of an "inhomogeneous Luttinger liquid" with non-constant Luttinger parameter $K(x)$. It turns out that, in this particularly simple setup, the Green's functions can be expressed analytically. Maslov and Stone [34] used a Lagrangian formulation and therefore wrote the action of the IGFF (8) — see Eq. (3) in their paper —; to our knowledge, this is the first time that action appeared in the literature. Maslov and Stone also derived a differential equation for the propagator (Eq. (6) in their paper) that is similar to the generalized Poisson equation from Sec. 2 below. The same model was studied by Fazio, Hekking and Khmelnitskii [36] in the context of thermal transport. However, the physical quantities studied in Refs. [34–36] were simply defined in terms of integrals of the propagator, so the authors did not have to push further the calculation of more general correlation functions.

About a decade later, in 2003, Gangardt and Shlyapnikov [37] had similar insights, and wrote the Hamiltonian of the inhomogeneous Luttinger liquid (see the equation above Eq. (12) in their paper), this time with the purpose of computing correlation functions of a 1d Bose gas trapped in a harmonic potential. They took the Luttinger liquid Hamiltonian [2], assumed that $K$ and $v$ were both position-dependent, and then used the Local Density Approximation (LDA) to fix these parameters. They extracted $K(x)$ and $v(x)$ from the Bethe ansatz solution of the homogeneous Lieb-Liniger model (see also Ref. [38] where LDA was used to calculate *local* correlation functions). This is exactly what we will do in Sec. 3 below. From there, they

derived an expansion of the boson field which, in principle, allows to compute correlation functions. The same logic was followed by Ghosh in 2006 [39] and by Citro et al. in 2008 [40]. Some of these results have been reviewed in Ref. [21] (section V.E).

The same kind of approach was also developed in the context of multi-component 1d Fermi gases. In 2003, following the spirit of [41], Recati et al. [42] investigated the spectrum and discussed experimental realizations of spinful ultra-cold Fermi gases; independently of [37], the authors assumed that the space-dependent parameters $K(x)$ and $v(x)$ could be fixed by LDA. This idea was later used by Liu et al. [43,44] to study the phase diagram of the 1d Hubbard model. As far as we are aware, this has not been explicitly used to compute correlation functions in this context.

The innovation of the present paper, compared to Refs. [34–40], is twofold. First, in Sec. 2 we discuss the IGFF and its correlation functions in full generality. To our knowledge, such a general and complete discussion has not appeared elsewhere, and it should be useful to some readers. Second, we believe that an important ingredient has been missed in Refs. [37,39,40], and that the results for correlation functions reported in those references are, in fact, not entirely correct. The reason is the following.

In general, local observables in a microscopic model $\hat{O}(x)$ (say, the Lieb-Liniger model) are related to field theory operators $\phi(x)$ only through non-universal coefficients $C$. To elaborate, observables $\hat{O}(x)$ are expected to have expansions of the form

$$\hat{O}(x) = \sum_j C_j^{(\hat{O})} \phi_j(x), \tag{1}$$

where the sum in the r.h.s. runs over all possible local operators $\phi_j$ in the field theory, and the non-universal coefficients $C_j^{(\hat{O})}$ are dimensionful numbers. As usual, such an expansion is to be understood as a statement about correlation functions: correlations functions in the microscopic model are related to the ones of the field theory, providing asymptotic expansions of the former in the limit where all the points are well separated,

$$\left\langle \hat{O}_1(x_1) \dots \hat{O}_n(x_n) \right\rangle_{\text{microsc.}} = \sum_{j_1,\dots,j_n} C_{j_1}^{(\hat{O}_1)} \dots C_{j_n}^{(\hat{O}_n)} \left\langle \phi_{j_1}(x_1) \dots \phi_{j_1}(x_n) \right\rangle_{\text{field th.}}.$$

In homogeneous systems, the non-universal coefficients merely contribute as global prefactors in the correlation functions (a useful and detailed discussion of those coefficients can be found in Refs. [46–49]). But, in inhomogeneous situations, those dimensionful coefficients $C_j^{(O)}$ are themselves position-dependent, $C_j^{(O)} \rightarrow C_j^{(O)}(x)$, so they have a crucial impact on the correlators. This point seems to have been overlooked in previous works, see Fig. 8 in App. B for a plot comparing our result to the case where these coefficients are omitted.

In this paper, we use LDA to fix those dimensionful coefficients. We illustrate this in Sec. 3 in the Lieb-Liniger model. The prefactors are extracted from form factors formulae derived in the 1990s by algebraic Bethe ansatz [50–52], see App. B for more information. The method is then checked against numerical results in the Lieb-Liniger model obtained from DMRG, using the C++ library ITensor [53], see App. D for details about the simulation. The agreement is quite impressive, as can be seen in Figs. 5, 6 and 7 below.

## 1.2 The underlying assumption: separation of scales

The approach we adopt in this paper is valid in the limit where the system exhibits *separation of scales*, see Fig. 1. This is the limit where the confining potential $V(x)$, and more generally all local thermodynamic quantities of the quantum gas — such as its particle density, energy

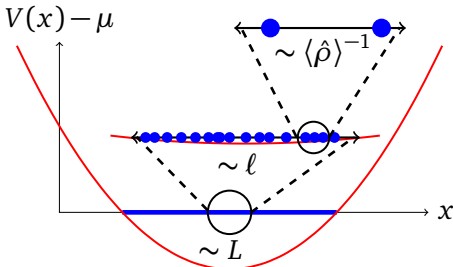

Figure 1: Cartoon illustrating the separation of scales in a trapped 1d gas. The typical length $L$ on which the local chemical potential $\mu(x) = \mu - V(x)$ varies is of the order of the total size of the system. This *macroscopic scale* is much larger than the *microscopic scale* corresponding to the inverse density $\langle \hat{\rho} \rangle^{-1}$. There exists a *mesoscopic scale* $\ell$ at which the system consists of fluid cells that are locally homogeneous, but still contain a very large number of particles.

density, momentum density, etc. — vary very slowly on the *microscopic scale*. That microscopic scale is naturally given by the inverse density $\langle \hat{\rho}(x) \rangle^{-1}$, so the condition that the density varies slowly reads

$$\langle \hat{\rho}(x) \rangle^{-1} \ll \left( \frac{|\partial_x \langle \hat{\rho}(x) \rangle|}{\langle \hat{\rho}(x) \rangle} \right)^{-1} .$$

The r.h.s. defines a *macroscopic scale $L$*, which is typically of the order of the length of the system. When the macroscopic scale is much larger than the microscopic one, there exists an intermediate — or *mesoscopic* — scale $\ell$ such that

$$\langle \hat{\rho} \rangle^{-1} \ll \ell \ll L. \tag{2}$$

Then a "mesoscopic fluid cell" of size $\ell$ is both homogeneous (because it is small compared to the scale $L$ at which inhomogeneity becomes important) and contains a thermodynamically large number of particles (because it is large compared to $\langle \hat{\rho} \rangle^{-1}$). This is the key assumption that underlies the Local Density Approximation used in Refs. [37–40], and more generally all hydrodynamic approaches [54] (LDA itself being nothing but a "hydrostatic" approach [41]). The assumption is of course also a requirement for any effective field theory approach, because the fields themselves are supposed to describe coarse-grained versions of the microscopic degrees of freedom, and this makes sense only if there exist locally homogeneous cells over which coarse-graining can be performed.

In Sec. 3, we will explain in detail what limit we take in the trapped Lieb-Liniger model, and we will see that separation of scales holds exactly in our setup. The method we explore in this paper (which extends the previous results of Refs. [34–40]) should then be interpreted as a way of writing asymptotic expansions of the correlation functions in the $N \to +\infty$ limit, including not only the leading order, but also the first few finite-$N$ corrections.

## 1.3 The effective Gaussian action

To conclude this introduction, we explain why the problem of a quantum gas of particles in a trap leads to the IGFF, defined by the action (8). The content of this subsection is very similar to arguments given in Refs. [16, 17]; we repeat those here only for completeness.

There are several ways of showing the connection between Luttinger liquids and the GFF (homogeneous or not). Here we give an argument that is particularly short and is a good introduction to Secs. 2 and 3. More standard introductions can be found for instance in Refs. [4, 5].

The argument consists of two steps.

**Mapping on configurations of a height field —**   in 1d, configurations of indistinguishable particles can be represented by an height function $h(x)$ via

$$\hat{\rho}(x) = \frac{1}{2\pi} \partial_x h(x), \tag{3}$$

namely $h(x)$ is a real-valued function that is piecewise constant and jumps by $2\pi$ at the position of a particle. It is defined only up to a constant shift, $h \rightarrow h + \text{const.}$ To calculate ground state correlation functions, it is useful to imagine that the system evolves in imaginary time, and focus on correlation functions at arbitrary points $(x, \tau)$ in spacetime, and then later specify that all points are taken at imaginary time $\tau = 0$. For instance, the two-point function of the height field would be

$$\left\langle h(x, \tau) h(x', \tau') \right\rangle = \lim_{\beta \to \infty} \frac{\text{tr}[e^{-(\beta - \tau)H} h(x) e^{-(\tau - \tau')H} h(x') e^{-\tau' H}]}{\text{tr}[e^{-\beta H}]}, \tag{4}$$

where $H$ is the Hamiltonian of the system, and $\beta$ is the inverse temperature, that one sends to zero. The fluctuating field $h(x, \tau)$ is then viewed as a function on 2d spacetime.

**Action for the height field —**   the second step consists in writing an action for the height field $h(x, \tau)$. Our choice for the action is guided by two basic observations. First, assuming that the underlying microscopic model is described by a local Hamiltonian $H$, the action should be local. Second, physical observables should be invariant under constant shifts $h \rightarrow h + \text{const.}$, so the action must also possess that symmetry. This leads us to the general form

$$S[h] = \int \mathcal{L}(\partial_x h, \partial_\tau h, \dots) \, dx \, d\tau, \tag{5}$$

where the dots stand for higher order derivatives. The Lagrangian density $\mathcal{L}$ cannot depend on $h(x, \tau)$ itself, only on its derivatives, because we are asking that it is invariant under constant shifts. Finally, assuming that the action $S[h]$ is minimized by a unique classical configuration of the field, $h_{\text{cl.}}$, we can expand to second order around that minimum,

$$S[h + h_{\text{cl.}}] - S[h_{\text{cl.}}] =$$
$$\frac{1}{2} \int \left[ \frac{\partial^2 \mathcal{L}}{\partial (\partial_x h)^2} (\partial_x h)^2 + 2 \frac{\partial^2 \mathcal{L}}{\partial (\partial_x h) \partial (\partial_\tau h)} (\partial_x h)(\partial_\tau h) + \frac{\partial^2 \mathcal{L}}{\partial (\partial_\tau h)^2} (\partial_\tau h)^2 \right] dx \, d\tau$$
$$+ \text{ higher order terms.} \tag{6}$$

In 2d, higher order terms have scaling dimensions larger than 2 and are RG irrelevant; we can therefore discard them. The only free parameters of the effective theory are then the three independent real components of the Hessian $\nabla^2 \mathcal{L}$ at $h = h_{\text{cl.}}$, which is a positive $2 \times 2$ symmetric matrix that typically depends on position. It is convenient to interpret the inverse of that matrix as an emergent metric on spacetime

$$g = \begin{pmatrix} \frac{\partial^2 \mathcal{L}}{\partial (\partial_x h)^2} & \frac{\partial^2 \mathcal{L}}{\partial (\partial_x h) \partial (\partial_\tau h)} \\ \frac{\partial^2 \mathcal{L}}{\partial (\partial_x h) \partial (\partial_\tau h)} & \frac{\partial^2 \mathcal{L}}{\partial (\partial_\tau h)^2} \end{pmatrix}^{-1}, \tag{7}$$

and to rewrite the Gaussian action as

$$\begin{aligned} \mathcal{S}[h] &= S[h + h_{\text{cl.}}] - S[h_{\text{cl.}}] \\ &= \frac{1}{8\pi} \int \frac{\sqrt{g} d^2 \mathrm{x}}{K(\mathrm{x})} g^{ij} \partial_i h \partial_j h, \end{aligned} \tag{8}$$

where $(\mathrm{x}^1, \mathrm{x}^2) = (x, \tau)$ and $\frac{1}{K(\mathrm{x})} = 4\pi \sqrt{\det(\nabla^2 \mathcal{L})}$.

   This is the action of the IGFF. It is the most general action for the height field $h$ that is both local and invariant under constant shifts. We now study this theory in greater detail.

# 2 The 2d Inhomogeneous Gaussian Free Field

This section is devoted to the 2d Inhomogeneous Gaussian Free Field, which is the mathematical object that underlies inhomogeneous Luttinger liquids. The IGFF is a rather straightforward generalization of the Gaussian Free Field, parametrized by a function $K : \Omega \to \mathbb{R}_{>0}$ that represents the position-dependent coupling strength in the action $\mathcal{S}[h] = \frac{1}{8\pi} \int_\Omega \frac{\sqrt{g} d^2 x}{K(x)} g^{ij} \partial_i h \partial_j h$. The usual GFF is recovered when the function $K$ is constant, simply by rescaling the height field $h \to \sqrt{K} h$. In other words, while the usual GFF depends only on the domain $\Omega$ and on the conformal class of the metric g [11], the IGFF also depends on the function $K$.

For simplicity, we work on a simply connected, open subset $\Omega \subset \mathbb{R}^2$. [Later, when we will apply the IGFF to inhomogeneous Luttinger liquids, $\Omega$ will be identified with spacetime.] Since every metric in 2d is conformally flat, and because the action (8) is invariant under Weyl transformations $g_{ij} \to e^{2\sigma} g_{ij}$, w.l.o.g. we can work in the Euclidean metric

$$g_{ij} = \delta_{ij}, \tag{9}$$

such that the action of the IGFF becomes

$$\mathcal{S}[h] = \frac{1}{8\pi} \int_\Omega \frac{d^2 x}{K(x)} \left( \nabla h(x) \right)^2 . \tag{10}$$

Here $h$ is a real-valued function on the closure $\overline{\Omega}$, with Dirichlet boundary conditions,

$$h(x) = 0, \qquad \text{if} \quad x \in \partial\Omega. \tag{11}$$

## 2.1 Propagator of the IGFF, generalized Poisson equation

Correlation functions can be defined as path integrals,

$$\langle h(x_1) \dots h(x_n) \rangle = \frac{\int [dh] e^{-\mathcal{S}[h]} h(x_1) \dots h(x_n)}{\int [dh] e^{-\mathcal{S}[h]}}, \tag{12}$$

and since the action $\mathcal{S}[h]$ is Gaussian, the connected part of all $n$-point correlations with $n \geq 3$ vanishes. The 1-point function also vanishes, because it is antisymmetric under $h \mapsto -h$. Thus, all the information about the IGFF is contained in the 2-point function. From the action $\mathcal{S}[h]$, one can derive a constraint on the 2-point function as follows,

$$
\begin{aligned}
0 &= \frac{\int [dh] \frac{\delta}{\delta h(x)} \left( e^{-\mathcal{S}[h]} h(x') \right)}{\int [dh] e^{-\mathcal{S}[h]}} \\
&= -\left\langle \frac{\delta \mathcal{S}[h]}{\delta h(x)} h(x') \right\rangle + \delta^{(2)}(x - x') \\
&= \frac{1}{4\pi} \left\langle \nabla_x \cdot \left[ \frac{1}{K(x)} \nabla_x h(x) \right] h(x') \right\rangle + \delta^{(2)}(x - x'),
\end{aligned}
$$

where we integrated by parts in the last line. Thus, the 2-point function is identified with the Green's function of a generalised Poisson operator $\nabla \cdot \frac{1}{K(x)} \nabla$, namely

$$\left\langle h(x) h(x') \right\rangle = -G^{\mathrm{D}}_{[K]}(x, x'), \tag{13}$$

where $G^{\mathrm{D}}_{[K]}(x, x')$ is symmetric under exchange of x and x', and solves the linear differential problem

$$
\begin{cases}
\nabla_x \cdot \dfrac{1}{K(x)} \nabla_x G^{\mathrm{D}}_{[K]}(x, x') = 4\pi \delta^2(x - x') \\[2mm]
G^{\mathrm{D}}_{[K]}(x, x') = 0 \quad \text{for} \quad x \in \partial\Omega.
\end{cases}
\tag{14}
$$

The superscript 'D' refers to the boundary conditions (Dirichlet), while the subscript $[K]$ emphasizes the fact that the IGFF is parameterized by the function $K : \Omega \to \mathbb{R}_{>0}$. Contrary to the GFF, where the Green's function is easily obtained by conformal mapping of the domain $\Omega$ onto the upper half-plane (leading to explicit formulas in a number of physically relevant problems), no such explicit expression is available in general for the IGFF. The Green's function of the generalized Poisson operator can, however, be efficiently calculated numerically.

We note that the generalised Poisson operator is well-known from classical electrostatics [55]: it appears in the equation satisfied by the electrostatic potential $V(\mathbf{x})$ in the presence of a spatially-varying dielectric constant $\varepsilon(\mathbf{x})$: $\nabla \cdot \varepsilon(\mathbf{x}) \nabla V(\mathbf{x}) = 0$. The analogy with electrostatics will be pushed further below.

In summary, $n$-point correlation functions $\langle h(\mathbf{x}_1) \dots h(\mathbf{x}_n) \rangle$ in the IGFF are all expressible in terms of the Green's function of a generalised Poisson operator; notice that the result is divergent when $\mathbf{x}_i \to \mathbf{x}_j$, because the Green's function has a logarithmic singularity.

In applications to inhomogeneous Luttinger liguids, we need a few additional results about the IGFF, which provide natural generalisations of the ones that are well known for the GFF. First, we need to deal with vertex operators, which requires that we make sense of correlation functions of several insertions of $h(\mathbf{x})$ at the same point. This is what we do in the next subsection. Second, we need to compactify the field $h$ (meaning that we must view $h$ as taking values in the circle $\mathbb{R}/2\pi\mathbb{Z}$ instead of the real line $\mathbb{R}$), which we do in subsequent subsections.

## 2.2 Correlations at equal points, regularized Green's function

As usual in field theory, one needs a regularization procedure to make sense of multiple insertions of the field $h(\mathbf{x})$ at the same point, $h^n(\mathbf{x})$, $n \geq 2$. This is provided by the *normal order*, noted $: h^n(\mathbf{x}) :$, which is conveniently defined as follows. For $n = 0$, $: 1 := 1$, and for $n = 1$,

$$: h(\mathbf{x}) : = h(\mathbf{x}), \tag{15a}$$

and then, by induction on $n$, one defines $: h^n(\mathbf{x}) :$ as

$$: h^n(\mathbf{x}) : = \lim_{\mathbf{x}' \to \mathbf{x}} \left[ : h^{n-1}(\mathbf{x}) : h(\mathbf{x}') + (n-1) K(\mathbf{x}) \log \left| \mathbf{x} - \mathbf{x}' \right|^2 : h^{n-2}(\mathbf{x}) : \right]. \tag{15b}$$

The second term is introduced to cancel the divergence of the Green's function, $G_{[K]}^{\mathrm{D}}(\mathbf{x}, \mathbf{x}') \simeq K(\mathbf{x}) \log \left| \mathbf{x} - \mathbf{x}' \right|^2$, when $\mathbf{x}' \to \mathbf{x}$. With that definition, the expectation value $\langle : h^n(\mathbf{x}) : \rangle$ is finite, and is equal to

$$\langle : h^n(\mathbf{x}) : \rangle = \frac{n!}{2^{\frac{n}{2}} \left( \frac{n}{2} \right)!} \left( -G_{[K]}^{\mathrm{D}}(\mathbf{x}) \right)^{\frac{n}{2}}, \qquad (n \text{ even}). \tag{16}$$

[If $n$ is odd, the expectation value vanishes because of the symmetry $h \mapsto -h$.] The function appearing in the r.h.s. of Eq. (16) is the *regularized Green's function*, defined as

$$G_{[K]}^{\mathrm{D}}(\mathbf{x}) = \lim_{\mathbf{x}' \to \mathbf{x}} \left[ G_{[K]}^{\mathrm{D}}(\mathbf{x}, \mathbf{x}') - K(\mathbf{x}) \log \left| \mathbf{x} - \mathbf{x}' \right|^2 \right]. \tag{17}$$

This regularized Green's function will appear many times in the following. Notice that we use almost the same notation as for the Green's function itself, $G_{[K]}^{\mathrm{D}}(\mathbf{x}, \mathbf{x}')$, but with a single argument instead of two.

## 2.3 Vertex operators, analogy with electric charges

Exponentials of the field $h(\mathbf{x})$ define *vertex operators*, as in the usual GFF,

$$\mathcal{V}_\alpha(\mathbf{x}) = :e^{i\alpha h(\mathbf{x})}: \ = \sum_{p\geq 0} \frac{(i\alpha)^p}{p!} :h^p(\mathbf{x}): \ .$$

Correlation functions of vertex operators can be computed directly from their definition, using Wick's theorem; this is a standard exercise of combinatorial nature which we leave to the reader. The result is

$$\left\langle \mathcal{V}_{\alpha_1}(\mathbf{x}_1)\dots\mathcal{V}_{\alpha_n}(\mathbf{x}_n)\right\rangle = \left(\prod_{p=1}^{n} e^{\frac{\alpha_p^2}{2}G_{[K]}^D(\mathbf{x}_p)}\right)\left(\prod_{1\leq i<j\leq n} e^{\alpha_i\alpha_j G_{[K]}^D(\mathbf{x}_i,\mathbf{x}_j)}\right). \tag{18}$$

In the literature, such vertex operators are sometimes referred to as *electric charges*, in analogy with 2d electrostatics [55]. A simple way of seeing the analogy is to interpret the expectation value of $i\nabla h(\mathbf{x})$, as an electric field $E(\mathbf{x})$ in the plane. [Here, the factor $i$ is cosmetic; it is inserted in order to cancel the one in the exponential that defines the vertex operator, such that the expectation value of $i\nabla h(\mathbf{x})$ is real.] In the presence of vertex operators, $i\nabla h(\mathbf{x})$ acquires a non-zero expectation value,

$$\begin{aligned}
E(\mathbf{x}) &= \frac{\left\langle i\nabla h(\mathbf{x})\mathcal{V}_{\alpha_1}(\mathbf{x}_1)\dots\mathcal{V}_{\alpha_n}(\mathbf{x}_n)\right\rangle}{\left\langle \mathcal{V}_{\alpha_1}(\mathbf{x}_1)\dots\mathcal{V}_{\alpha_n}(\mathbf{x}_n)\right\rangle} \\
&= \nabla_{\mathbf{x}}\left[\left(\frac{\partial}{\partial\alpha}\log\left\langle \mathcal{V}_\alpha(\mathbf{x})\mathcal{V}_{\alpha_1}(\mathbf{x}_1)\dots\mathcal{V}_{\alpha_n}(\mathbf{x}_n)\right\rangle\right)_{\alpha=0}\right] \\
&= \sum_{j=1}^{n}\alpha_j\nabla_{\mathbf{x}}G_{[K]}^D(\mathbf{x},\mathbf{x}_j).
\end{aligned} \tag{19}$$

Thus, $E(\mathbf{x})$ satisfies Maxwell's equations in a medium with dielectric constant $\varepsilon(\mathbf{x}) = 1/K(\mathbf{x})$, with pointlike electric charges at positions $\mathbf{x}_j$,

$$\begin{cases}
\nabla\cdot\dfrac{1}{K(\mathbf{x})}E(\mathbf{x}) &= \displaystyle\sum_{j=1}^{n}4\pi\alpha_j\delta^{(2)}(\mathbf{x}-\mathbf{x}_j), \\
\nabla\times E(\mathbf{x}) &= 0.
\end{cases} \tag{20}$$

The first equation is the Gauss law for the displacement field $\varepsilon(\mathbf{x})E(\mathbf{x})$, and the second one is the Faraday law (in the absence of magnetic flux through the plane) which is automatically satisfied here because $E(\mathbf{x})$ is a gradient.

In fact, the logarithm of the correlation function (18) is nothing but the electrostatic energy of those pointlike electric charges, in the domain $\Omega$ with a local dielectric constant $\varepsilon(\mathbf{x}) = 1/K(\mathbf{x})$, surrounded by a perfect conductor (corresponding to the Dirichlet boundary condition),

$$\frac{1}{8\pi}\int\frac{d^2\mathbf{x}}{K(\mathbf{x})}|E(\mathbf{x})|^2 = \sum_{p=1}^{n}\frac{\alpha_p^2}{2}G_{[K]}^D(\mathbf{x}_p) + \sum_{i<j}\alpha_i\alpha_j G_{[K]}^D(\mathbf{x}_i,\mathbf{x}_j). \tag{21}$$

The second term is of course the Coulomb interaction for all the pairs of particles, while the first term is the electrostatic energy of each independent particle that arises from its interaction with the medium and with the perfect conductor at the boundary. Notice that the integral in the l.h.s. needs to be properly regularized to recover the regularized Green's function in that first term.

## 2.4 Compactification of the height field, magnetic operators

So far, we have assumed that the height field $h(\mathrm{x})$ was real-valued. From now on, we compactify the target space, and $h(\mathrm{x})$ is viewed as a point in $\mathbb{R}/2\pi\mathbb{Z}$ instead of $\mathbb{R}$. This compactification has two important consequences on the theory.

The first consequence is the quantization of electric charges: in order to be well-defined, the vertex operator $\mathcal{V}_\alpha(\mathrm{x}) = \, : e^{i\alpha h(\mathrm{x})} :$ must be invariant under $h \to h + 2\pi$. This implies that $\alpha$ is an integer.

The second consequence is the appearance of a new type of local operator $\mathcal{O}_\beta(\mathrm{y})$, representing a puncture at point $\mathrm{y} \in \Omega$, around which the field $h(\mathrm{x})$ has non-zero winding: $h(\mathrm{x})$ jumps by $2\pi\beta$, for some integer $\beta$, when x is dragged around the puncture counterclockwise. In other words,

$$\oint_{C_\mathrm{y}} d\mathrm{x} \cdot \nabla h(\mathrm{x}) = 2\pi\beta, \tag{22}$$

where $C_\mathrm{y}$ is a small oriented contour enclosing the point y. This identity holds when inserted inside correlation functions, e.g.

$$\oint_{C_{\mathrm{y}_j}} d\mathrm{x} \cdot \left\langle \nabla h(\mathrm{x}) \mathcal{O}_{\beta_1}(\mathrm{y}_1) \dots \mathcal{O}_{\beta_m}(\mathrm{y}_m) \right\rangle = 2\pi\beta_j \left\langle \mathcal{O}_{\beta_1}(\mathrm{y}_1) \dots \mathcal{O}_{\beta_m}(\mathrm{y}_m) \right\rangle. \tag{23}$$

Due to Dirichlet boundary conditions that impose that the contour integral along the boundary $\partial\Omega$ vanishes, $\oint_{\partial\Omega} d\mathrm{x} \cdot \nabla h(\mathrm{x}) = 0$, the set of operators $\mathcal{O}_{\beta_1}(\mathrm{y}_1), \dots, \mathcal{O}_{\beta_m}(\mathrm{y}_m)$ inserted inside a non-vanishing correlator must satisfy the neutrality condition

$$\beta_1 + \cdots + \beta_m = 0. \tag{24}$$

The operators $\mathcal{O}_\beta$ are often called "magnetic operators" in the literature. Again, this is an explicit reference to the electrostatic analogy. Indeed, the equations satisfied by the "electric field" $E(\mathrm{x})$, namely the expectation value of $\nabla h(\mathrm{x})$ (here we drop the cosmetic $i$ from the previous subsection, because the expectation value of $\nabla h(\mathrm{x})$ is real) with insertions of those operators,

$$E(\mathrm{x}) = \frac{\left\langle \nabla h(\mathrm{x}) \mathcal{O}_{\beta_1}(\mathrm{y}_1) \dots \mathcal{O}_{\beta_m}(\mathrm{y}_m) \right\rangle}{\left\langle \mathcal{O}_{\beta_1}(\mathrm{y}_1) \dots \mathcal{O}_{\beta_m}(\mathrm{y}_m) \right\rangle}, \tag{25}$$

are:

$$\begin{cases} \nabla \cdot \dfrac{1}{K(\mathrm{x})} E(\mathrm{x}) & = & 0, \\[2mm] \nabla \times E(\mathrm{x}) & = & \displaystyle\sum_{j=1}^{m} 2\pi\beta_j \, \delta^{(2)}\left(\mathrm{x} - \mathrm{y}_j\right). \end{cases} \tag{26}$$

The first constraint is the equation of motion for $h(\mathrm{x})$ derived from the action (10). Again, we view it as the Gauss law in a medium with dielectric constant $\varepsilon(\mathrm{x})$, this time without electric charges. The second is just a rewriting of Eq. (23) using Stokes' formula, and we regard it as the Faraday law, imagining that the plane is transpierced by infinitely thin, constantly increasing, magnetic fluxes at positions $\mathrm{y}_j$.

## 2.5 Correlation functions of magnetic operators from electric-magnetic duality

We now turn to the calculation of correlators of magnetic operators. Again, such correlators are defined as a path integral

$$\left\langle \mathcal{O}_{\beta_1}(\mathrm{y}_1) \dots \mathcal{O}_{\beta_m}(\mathrm{y}_m) \right\rangle = \frac{\int_{\text{defects}} [dh_\mathrm{d}] e^{-\mathcal{S}[h_\mathrm{d}]}}{\int [dh] e^{-\mathcal{S}[h]}}, \tag{27}$$

where the path integral in the numerator is over functions $h_d$ from the punctured domain $\Omega \setminus \{y_1, \ldots, y_m\}$ to $\mathbb{R}/2\pi\mathbb{Z}$ that have the correct winding $\beta_j$ around each puncture $y_j$,

$$\oint_{C_{y_j}} d\mathbf{x} \cdot \nabla h_d = 2\pi\beta_j. \tag{28}$$

We refer to those as "height configurations with defects". In this subsection (and only here), we use a subscript 'd' for configurations with defects. The denominator in Eq. (27) is the path integral on configurations without defects, namely the partition function of the IGFF on $\Omega$.

The numerator can be evaluated by separating the configurations with defects into a classical part that satisfies the equation of motion, and a quantum, or fluctuating, part:

$$h_d(\mathbf{x}) = h_d^0(\mathbf{x}) + h(\mathbf{x}). \tag{29}$$

Since both $h_d(\mathbf{x})$ and $h_d^0(\mathbf{x})$ satisfies the constraint (28), $h(\mathbf{x})$ is a single-valued real function on $\Omega$. Moreover, since $h_d^0(\mathbf{x})$ is assumed to satisfy the equation of motion, the action splits,

$$\mathcal{S}[h_d] = \mathcal{S}[h_d^0] + \mathcal{S}[h]. \tag{30}$$

By a trivial change of variables $h_d(\mathbf{x}) \mapsto h(\mathbf{x})$, the path integral in the numerator of (27) becomes an integral of the fluctuating part $h(\mathbf{x})$ which cancels the one in the denominator. So the correlation function (27) boils down to

$$\langle \mathcal{O}_{\beta_1}(y_1) \ldots \mathcal{O}_{\beta_m}(y_m) \rangle = e^{-\mathcal{S}[h_d^0]}, \tag{31}$$

and the remaining task is to calculate the integral

$$\mathcal{S}[h_d^0] = \frac{1}{8\pi} \int_\Omega \frac{d^2\mathbf{x}}{K(\mathbf{x})} (\nabla h_d^0(\mathbf{x}))^2, \tag{32}$$

where $h_d^0(\mathbf{x})$ satisfies the constraint (28), the equation of motion $\nabla \cdot \frac{1}{K(\mathbf{x})} \nabla h_d^0(\mathbf{x}) = 0$, and possesses Dirichlet boundary conditions.

The electrostatic analogy provides an elegant way of calculating that integral. Indeed, the integral is nothing but the electrostatic energy $\frac{1}{8\pi} \int d^2\mathbf{x}\, \varepsilon(\mathbf{x}) |E(\mathbf{x})|^2$ for the electric field $E(\mathbf{x}) = \nabla h_d^0(\mathbf{x})$ created by constantly increasing fluxes that pierce the plane. If we could trade those magnetic fluxes for pointlike electric charges, then the answer would be given by Eq. (21).

This can be done by electric-magnetic duality. If we define a new field $\tilde{E}$ with components

$$\begin{pmatrix} \tilde{E}_1 \\ \tilde{E}_2 \end{pmatrix} = \frac{1}{2K} \begin{pmatrix} E_2 \\ -E_1 \end{pmatrix}, \tag{33}$$

then we see that the constraints (26) for $E$, with dielectric constant $1/K$, are turned into the constraints (20) for $\tilde{E}$ with dielectric constant $1/\tilde{K} = 4K$. [In Eq.(33), we introduced an extra factor 2 such that the Green's function is defined in its standard form with a factor $4\pi$.] Now, we can apply formula (21):

$$\begin{aligned} \mathcal{S}[h_d] &= \frac{1}{8\pi} \int \frac{d^2\mathbf{x}}{K} |E|^2 = \frac{1}{8\pi} \int \frac{d^2\mathbf{x}}{\tilde{K}} |\tilde{E}|^2 \\ &= \sum_{p=1}^n \frac{\beta_p^2}{2} G_{[\tilde{K}]}^N(y_p) + \sum_{i<j} \beta_i \beta_j G_{[\tilde{K}]}^N(y_i, y_j). \end{aligned} \tag{34}$$

In the last line, notice that we have replaced the superscript 'D' by 'N'. This is because Dirichlet boundary conditions are dual to Neumann boundary conditions. To see this, one can think of $E$ as $\nabla h_{\mathrm{d}}$, and of $\tilde{E}$ as the gradient $\nabla \tilde{h}$ of some other function $\tilde{h}$. Because $h_{\mathrm{d}} = 0$ at the boundary $\partial \Omega$, the component $E_{\parallel}$ that is tangential to the boundary vanishes. Since $\tilde{E}$ is obtained from a $\pi/2$-rotation of $E$, this implies that the normal component $\tilde{E}_{\perp}$ vanishes. Hence, the dual field $\tilde{h}$ has Neumann boundary conditions, instead of Dirichlet.

In summary, the result for the correlation function of magnetic operators is

$$\left\langle \mathcal{O}_{\beta_1}(\mathrm{y}_1) \dots \mathcal{O}_{\beta_m}(\mathrm{y}_m) \right\rangle = \left( \prod_{p=1}^{m} e^{\frac{\beta_p^2}{2} G_{[1/4K]}^{\mathrm{N}}(\mathrm{y}_p)} \right) \left( \prod_{1 \le i < j \le m} e^{\beta_i \beta_j G_{[1/4K]}^{\mathrm{N}}(\mathrm{y}_i, \mathrm{y}_j)} \right), \qquad (35)$$

where the Green's function (as well as its regularised version, defined exactly as in Eq. (17) above) is the one of the generalized Poisson operator $\nabla \cdot 4K\nabla$, with Neumann boundary conditions. This Green's function $G_{[1/4K]}^{\mathrm{N}}(\mathrm{y}, \mathrm{y}')$ is symmetric under exchange of $\mathrm{y}$ and $\mathrm{y}'$, and it solves the linear differential problem

$$\begin{cases} \nabla_{\mathrm{y}} \cdot 4K(\mathrm{y}) \nabla_{\mathrm{y}} G_{[1/4K]}^{\mathrm{N}}(\mathrm{y}, \mathrm{y}') = 4\pi \delta^2(\mathrm{y} - \mathrm{y}') - \frac{4\pi}{\mathrm{Vol}(\Omega)}, \\[2mm] \int_{\Omega} d^2\mathrm{y}\, G_{[1/4K]}^{\mathrm{N}}(\mathrm{y}, \mathrm{y}') = 0, \\[2mm] \hat{n}_{\mathrm{y}} \cdot \nabla_{\mathrm{y}} G_{[1/4K]}^{\mathrm{N}}(\mathrm{y}, \mathrm{y}') = 0 \quad \text{for} \quad \mathrm{y} \in \partial\Omega, \end{cases} \qquad (36)$$

with $\hat{n}_{\mathrm{y}}$ the unit vector normal to the boundary at $\mathrm{y} \in \partial\Omega$. The term $4\pi/\mathrm{Vol}(\Omega)$ in the first equation, as well as the second equation that imposes zero mean value, both come from the fact that the generalized Poisson operator $\nabla \cdot 4K\nabla$ with Neumann boundary conditions possesses a zero mode: it annihilates any constant function on $\Omega$. The second equation is then imposed to restrict the problem to the subspace of functions on $\Omega$ that have zero mean value. On that subspace, $\nabla \cdot 4K\nabla$ is invertible. The Green's function is then defined as the operator inverse on that subspace, which is what is expressed by the first equation, where both the l.h.s. and r.h.s. have zero mean value.

## 2.6 Mixed electric-magnetic correlators, the mixed function $F_{[K,1/4K]}^{\mathrm{D,N}}$

In some applications of the IGFF, one expects that we will need correlation functions involving both electric and magnetic operators. Once again, the electrostatic analogy provides a convenient way of evaluating such "mixed" correlators. Indeed, the result must take the form

$$\begin{aligned} \left\langle \mathcal{V}_{\alpha_1}(\mathrm{x}_1) \dots \mathcal{V}_{\alpha_n}(\mathrm{x}_n) \mathcal{O}_{\beta_1}(\mathrm{y}_1) \dots \mathcal{O}_{\beta_m}(\mathrm{y}_m) \right\rangle = \\ \left( \prod_{p=1}^{n} e^{\frac{\alpha_p^2}{2} G_{[K]}^{\mathrm{D}}(\mathrm{x}_p)} \prod_{1 \le i < j \le n} e^{\alpha_i \alpha_j G_{[K]}^{\mathrm{D}}(\mathrm{x}_i, \mathrm{x}_j)} \right) \\ \times \left( \prod_{q=1}^{m} e^{\frac{\beta_q^2}{2} G_{[1/4K]}^{\mathrm{N}}(\mathrm{y}_q)} \prod_{1 \le i < j \le m} e^{\beta_i \beta_j G_{[1/4K]}^{\mathrm{N}}(\mathrm{y}_i, \mathrm{y}_j)} \right) \\ \times \left( \prod_{k=1}^{n} \prod_{l=1}^{m} e^{i \alpha_k \beta_l F_{[K,1/4K]}^{\mathrm{D,N}}(\mathrm{x}_k, \mathrm{y}_l)} \right), \quad (37) \end{aligned}$$

such that its logarithm is the total electrostatic energy of a configuration of $n$ pointlike electric charges and $m$ punctures with insertions of fluxes. This total energy is a sum of $n + \frac{n(n-1)}{2} + m + \frac{m(m-1)}{2} + nm$ terms. Each of the first $n$ terms is the Coulomb energy of a single electric charge at position $\mathrm{x}_p$ in $\Omega$, the next $\frac{n(n-1)}{2}$ terms are the Coulomb energies of

each pair of electric charges. Similarly, $m$ terms are the energy of each individual flux insertion, and there are $\frac{m(m-1)}{2}$ terms for each pair of those. We have already encountered all those terms in previous subsections. The new $n \times m$ terms here are the ones that correspond to the energy of an electric charge at position $x_k$ in the electrostatic potential created by a magnetic flux inserted at $y_l$.

This potential, which we call $F^{D,N}_{[K,1/4K]}(x_k, y_l)$, is a function of x and y with value in $\mathbb{R}/2\pi\mathbb{Z}$ that satisfies a number of constraints, which we detail now. First, we need to choose a continuous function $f : \partial\Omega \to [0, 2\pi]$ with winding number one: $\oint_{\partial\Omega} dx \cdot \nabla f(x) = 2\pi$. Then $F^{D,N}_{[K,1/4K]}(x, y)$ is defined as the solution to the problem

$$
\begin{cases}
\nabla_x \cdot \dfrac{1}{K(x)} \nabla_x F^{D,N}_{[K,1/4K]}(x, y) &=\ 0 \quad \text{if}\quad x \in \Omega \setminus \{y\}, \\
F^{D,N}_{[K,1/4K]}(x, y) &=\ f(x) \quad \text{if}\quad x \in \partial\Omega.
\end{cases}
\tag{38}
$$

Notice that, as a consequence,

$$
\oint_{C_y} dx \cdot \nabla_x F^{D,N}_{[K,1/4K]}(x, y) = 2\pi
\tag{39}
$$

for any contour $C_y$ that encircles y.

It is important to stress that, while $F^{D,N}_{[K,1/4K]}(x, y)$ depends on the choice of the function $f$, the correlation function (37) does not. Indeed, imagine that we have two functions $f_1$ and $f_2$ with winding number one, and that we look at the corresponding $F_1(x, y)$ and $F_2(x, y)$ defined by Eqs. (38). Then $F_1(x, y_l) - F_2(x, y_l)$ is a continuous function for $x \in \Omega$, with no winding anywhere, that satisfies $\nabla \cdot \frac{1}{K} \nabla[F_1 - F_2] = 0$, with $F_1 - F_2 = f_1 - f_2$ along the boundary. Then, summing over $l$ from 1 to $m$ and using the neutrality condition (24), one sees that $\sum_{l=1}^m \beta_l [F_1(x, y_l) - F_2(x, y_l)]$ is a function that is annihilated by $\nabla \cdot \frac{1}{K} \nabla$, with boundary conditions $\sum_{l=1}^m \beta_l [F_1(x, y_l) - F_2(x, y_l)] = 0$. Thus, it has to vanish everywhere. So the correlation function (37) is independent of the choice of $f$ as claimed.

It is interesting to note that, when viewed as a function of y, $F^{D,N}_{[K,1/4K]}(x, y)$ satisfies a set of constraints that are dual to Eqs. (38):

$$
\begin{cases}
\nabla_y \cdot 4K(y) \nabla_y F^{D,N}_{[K,1/4K]}(x, y) &=\ 0, \\
\oint_{C_x} dy \cdot \nabla_y F^{D,N}_{[K,1/4K]}(x, y) &=\ -2\pi \\
\hat{n}_y \cdot \nabla_y F^{D,N}_{[K,1/K]}(x, y) &=\ \hat{t}_y \cdot \nabla f(y) \quad \text{if}\quad y \in \partial\Omega.
\end{cases}
\tag{40}
$$

where $\hat{n}_y$ and $\hat{t}_y$ are two unit vectors respectively normal and tangent to the boundary $\partial\Omega$ at position y. This is more easily seen by considering a discrete version of the compactified IGFF, in analogy with lattice electrostatics, see App. C.

## 2.7 Mixed electric-magnetic operators

Finally, it might also be convenient to deal directly with vertex operators that possess both an electric and a magnetic charge. The latter are obtained when one fuses an electric operator with a magnetic one, meaning that one takes the limit $x, y \to z$ in correlation functions involving $\mathcal{O}_\beta(y)$ and $\mathcal{V}_\alpha(x)$. It is therefore convenient to introduce a new notation for vertex operators that carry both an electric and a magnetic charge:

$$
\mathcal{V}_{\alpha,\beta}(z),
\tag{41}
$$

with two indices for the two charges, such that the previous "pure electric" or "pure magnetic" operators correspond to $\mathcal{V}_\alpha(x) = \mathcal{V}_{\alpha,0}(x)$ and $\mathcal{O}_\beta(y) = \mathcal{V}_{0,\beta}(y)$ respectively. The correlation function of such operators can be obtained by taking $m = n$ and $x_i, y_i \to z_i$ in Eq. (37). The result is

$$
\left\langle \mathcal{V}_{\alpha_1,\beta_1}(z_1) \ldots \mathcal{V}_{\alpha_n,\beta_n}(z_n) \right\rangle =
$$
$$
\left( \prod_{p=1}^{n} e^{\frac{\alpha_p^2}{2} G_{[K]}^{\mathrm{D}}(z_p) + \frac{\beta_p^2}{2} G_{[1/4K]}^{\mathrm{N}}(z_p) + i\alpha_p\beta_p F_{[K,1/4K]}^{\mathrm{D,N}}(z_p)} \right)
$$
$$
\times \left( \prod_{i \neq j} e^{\frac{\alpha_i\alpha_j}{2} G_{[K]}^{\mathrm{D}}(x_i,x_j) + \frac{\beta_i\beta_j}{2} G_{[1/4K]}^{\mathrm{N}}(y_i,y_j) + i\alpha_i\beta_j F_{[K,1/4K]}^{\mathrm{D,N}}(x_i,y_j)} \right). \quad (42)
$$

Here the regularized function $F_{[K,1/4K]}^{\mathrm{D,N}}(z)$ is defined as

$$
F_{[K,1/4K]}^{\mathrm{D,N}}(z) = \lim_{x,y \to z} \left[ F_{[K,1/4K]}^{\mathrm{D,N}}(x,y) - \arg(x-y) \right], \quad (43)
$$

where $\arg(x)$ is the argument of the complex number $x^1 + ix^2$ made out of the coordinates $x = (x^1, x^2)$. It is easy to see that this definition is compatible with the short-distance behavior of the function $F_{[K,1/4K]}^{\mathrm{D,N}}(z, z')$ that is imposed by Eq. (39).

This concludes this section on the (compactified) IGFF. Formula (42) for the correlation functions of mixed electric-magnetic vertex operators is all we need, since all correlation functions of local observables can be obtained from those. Thus, all correlation functions in the (compactified) IGFF can be expressed in terms of two Green's functions $G_{[K]}^{\mathrm{D}}$ and $G_{[1/4K]}^{\mathrm{N}}$ of two mutually dual generalized Poisson operators, and a third "mixed" function $F_{[K,1/4K]}^{\mathrm{D,N}}$, as well as their regularizations.

# 3 Application to the Lieb-Liniger model in a trap

We now turn to the problem of calculating correlation functions of trapped 1d Bose gases. This will illustrate how the machinery of the IGFF developed in Sec. 2 is useful in practice.

We focus on the Lieb-Liniger model of spinless bosons with repulsive delta interaction, defined by the Hamiltonian

$$
H = \int dx \left[ \hat{\Psi}^\dagger(x) \left( -\frac{\hbar^2}{2m} \partial_x^2 - \mu + V(x) \right) \hat{\Psi}(x) + \frac{\hbar\bar{g}}{2} \hat{\Psi}^{\dagger 2}(x) \hat{\Psi}^2(x) \right], \quad (44)
$$

where $\hat{\Psi}^\dagger(x)$ ($\hat{\Psi}(x)$) is the boson creation (annihilation) operator that satisfies the canonical commutation relation $[\hat{\Psi}(x), \hat{\Psi}^\dagger(x')] = \delta(x - x')$, $m$ is the mass of a boson, $g = \hbar\bar{g} > 0$ is the interaction strength, $\mu$ is the chemical potential and $V(x)$ is a trapping potential. There are two main reasons for focusing on this Hamiltonian: it is the model that is experimentally relevant to describe Bose gases through the whole range of repulsion strength in one dimension [56], and, in the homogeneous case $V(x) = 0$, it is exactly solvable by Bethe Ansatz (for an introduction to the Bethe Ansatz solution of the Lieb-Liniger model, see e.g. Ref. [57]).

Throughout this section, we consider that $m$, $\mu$, $V(x)$ and $\bar{g}$ are fixed parameters, and we focus on the limit $\hbar \to 0$. Our goal is to study correlation functions in the ground state of $H$, and to understand how to get the first few terms of their asymptotic expansion in $\hbar$ in that limit.

### 3.1 The limit $\hbar \to 0$

Taking the limit $\hbar \to 0$ while keeping all other parameters fixed is a particularly convenient way of taking the thermodynamic limit $N \to +\infty$. The reason is the following.

In the homogeneous case $V(x) = 0$, dimensional analysis shows that the particle density in the ground state must take the form

$$\langle \hat{\rho}(x) \rangle = \left\langle \hat{\Psi}^{\dagger}(x) \hat{\Psi}(x) \right\rangle = \frac{\sqrt{m\mu}}{\hbar} F\left( \bar{g}\sqrt{m/\mu} \right), \tag{45}$$

for any positive value of the chemical potential $\mu > 0$, where $F(.)$ is some function that can be calculated from Bethe Ansatz. Thus, at least in the homogeneous case, the density of particles diverges as $1/\hbar$ when $\hbar \to 0$.

Then, in the inhomogeneous case, one can rely on the following self-consistent argument. Assuming that the density of particles is sufficiently large at each point where the local chemical potential $\mu(x) = \mu - V(x)$ is positive, one can rely on separation of scales, see Fig. 1. Under this assumption, the density is

$$\langle \hat{\rho}(x) \rangle \underset{\hbar \to 0}{=} \rho_{\mathrm{LDA}}(x) := \frac{\sqrt{m\mu(x)}}{\hbar} F\left( \bar{g}\sqrt{m/\mu(x)} \right). \tag{46}$$

This is the Local Density Approximation. It shows that the density locally diverges as $1/\hbar$ at every point where $\mu - V(x) > 0$, thus separation of scales (see Fig. 1) becomes exact in the limit $\hbar \to 0$.

Since the total number of particles is the integral of the density $\langle \hat{\rho}(x) \rangle$ over the region where $\mu - V(x) > 0$, it is clear that $N \propto 1/\hbar$, so that limit is a thermodynamic limit, as claimed. Importantly, in our setup, the local dimensionless parameter

$$\gamma(x) := \frac{m\bar{g}}{\hbar \rho_{\mathrm{LDA}}(x)} \tag{47}$$

stays finite as $\hbar \to 0$. [This is in contrast with other possible ways of taking the thermodynamic limit (in particular, if one kept $g = \hbar \bar{g}$ fixed, instead of $\bar{g}$) where the dimensionless interaction parameter $\gamma$ could diverge.]

### 3.2 Fixing $K(x)$ and g(x) from LDA

For simplicity, we now assume that the domain where $\mu(x) = \mu - V(x)$ is positive is a single interval, which we take to be symmetric around the origin, $[-R, R]$, with $2R$ the total size of the boson cloud in the limit $\hbar \to 0$. To calculate ground state correlations, we then need to consider the IGFF defined in the spacetime domain $(x, \tau) \in \Omega := [-R, R] \times \mathbb{R}$. Importantly, in the ground state of the trapped gas, the density of particles vanishes at the edges, which imposes some boundary conditions on the height field $h(x, \tau)$. To see what they are, let us look back at the definition (3).

In Sec. 1.3, the effective Gaussian action $\mathcal{S}[h]$ was obtained by expanding $h$ around a classical configuration $h_{\mathrm{cl.}}$. It means that $h(x, \tau)$ is just the fluctuating part of the height function. So now, the definition (3) only makes sense if we invert it in the following way

$$h(x) = 2\pi \int_{-R}^{x} du \left[ \hat{\rho}(u) - \langle \hat{\rho}(u) \rangle \right], \tag{48}$$

which satisfies $\langle h(x) \rangle = 0$. But, since the total number of particles $N$ is fixed in the interval $[-R, R]$, this necessarily imposes Dirichlet boundary conditions,

$$h(-R, \tau) = h(R, \tau) = 0. \tag{49}$$

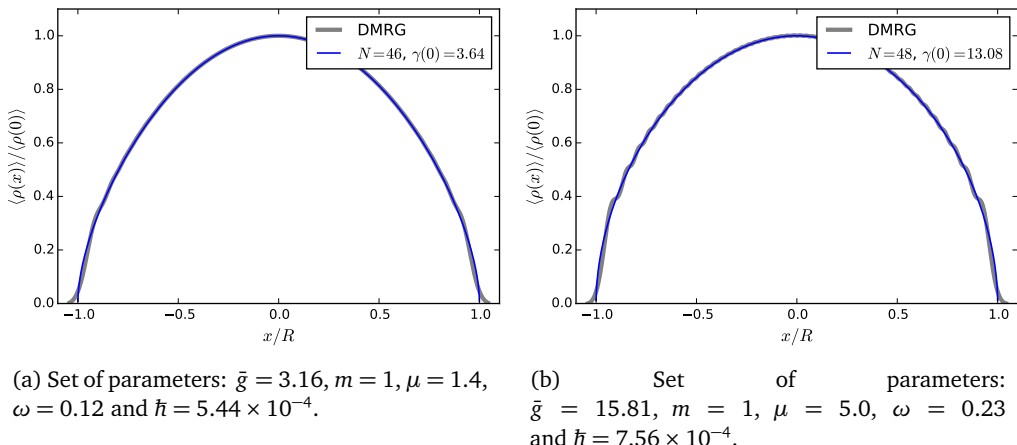

(a) Set of parameters: $\bar{g} = 3.16$, $m = 1$, $\mu = 1.4$, $\omega = 0.12$ and $\hbar = 5.44 \times 10^{-4}$.

(b) Set of parameters: $\bar{g} = 15.81$, $m = 1$, $\mu = 5.0$, $\omega = 0.23$ and $\hbar = 7.56 \times 10^{-4}$.

Figure 2: Density profiles in a harmonic potential $V(x) = \frac{1}{2}m\omega^2 x^2$. The profile $\rho_{\mathrm{LDA}}(x)$ obtained from the Local Density Approximation — see Eq. (46) and Ref. [41] — is compared to the exact profile obtained numerically from DMRG. The two sets of parameters shown here give rise to a different value of the dimensionless interacting parameter $\gamma(x)$. The minimum of $\gamma(x)$ is reached at the center of the trap $x = 0$.

Now, to apply the formalism of Sec. 2 to the trapped Lieb-Liniger gas, we need to fix the Luttinger parameter $K$ and the (conformal class of the) metric g on the domain $\Omega = [-R, R] \times \mathbb{R}$. To do this, we rely once again on separation of scales, and we use the exact solution from Bethe Ansatz that is available in the homogeneous case.

Thanks to separation of scales, we can imagine that we focus first on correlation functions within a single mesoscopic fluid cell, see Fig. 1. The mesoscopic cell is homogeneous and contains a thermodynamically large number of particles, so the correlation functions must be exactly the same as the ones of the homogeneous system in the thermodynamic limit. But, in the homogeneous problem, both $K$ and g are known, and the metric is simply (with $\mathrm{x} = (x, \tau)$)

$$
\begin{aligned}
ds^2 &= g_{ij} dx^i dx^j \\
&= dx^2 + v^2 d\tau^2.
\end{aligned}
\tag{50}
$$

By dimensional analysis, the effective velocity $v$ of gapless excitations above the ground state is of the form $v = v_{\mathrm{F}} f_1(\gamma)$ for some function $f_1$, where $v_{\mathrm{F}} = \sqrt{2\mu/m}$ is the Fermi velocity. Similarly, the (dimensionless) Luttinger parameter $K$ is of the form $f_2(\gamma)$. These functions $f_1$ and $f_2$ are known from Bethe Ansatz; they are plotted in Fig. 3.

This fixes the metric g and the parameter $K$ within each mesoscopic cell. Then, of course, the action of the IGFF in the entire domain $\Omega = [-R, R] \times \mathbb{R}$ is determined.

So, to sum up, we know what field theory needs to be solved: it is the IGFF in the metric (50), with a velocity $v$ and a Luttinger parameter $K$ that both depend on the position $x$ through the local density $\rho_{\mathrm{LDA}}(x)$ and the local dimensionless interaction parameter $\gamma(x)$. Correlation functions can thus be expressed in terms of the Green's functions $G^{\mathrm{D}}_{[K]}$ and $G^{\mathrm{N}}_{[1/4K]}$ defined in Sec. 2, which are efficiently calculated numerically.

We conclude this subsection with a short remark about the coordinate system. In Sec. 2, we relied on a system of isothermal coordinates to simplify the expressions associated with the differential operators — the generalized Poisson operators — whose Green's functions appear in the IGFF correlators. Here, a system of isothermal coordinates is readily available [13].

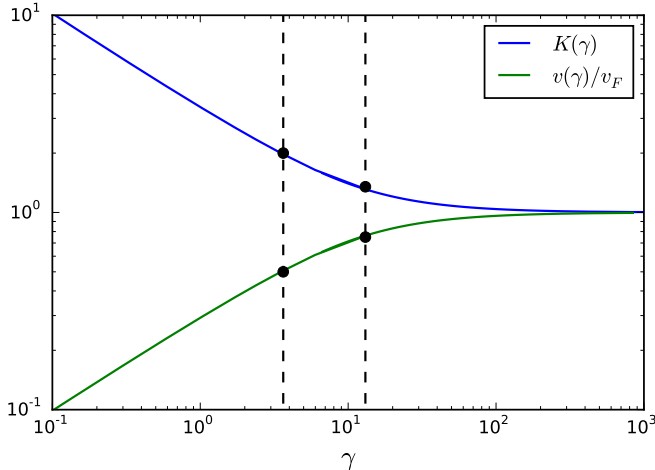

Figure 3: The Luttinger parameter $K$ and the velocity of gapless excitations $v$ (here divided by the Fermi velocity $v_{\mathrm{F}} = \sqrt{2\mu/m}$) are functions of the dimensionless interaction parameter $\gamma$ in the Lieb-Liniger model. Galilean invariance implies that $K(\gamma)v(\gamma)/v_{\mathrm{F}} = 1$, see Ref. [2]. The two dashed lines correspond to the two sets of parameters for which we provide DMRG checks in this paper (see Fig.2); the corresponding interaction parameter at the center of the trap are $\gamma(0) = 3.64$ and $\gamma(0) = 13.08$.

Indeed, one can stretch the spatial coordinate $x$ according to

$$\tilde{x} = \int_0^x \frac{\mathrm{d}u}{v(u)}, \tag{51}$$

such that $\tilde{x} \in [-\tilde{R}, \tilde{R}]$ with $\tilde{R} = \int_0^R \frac{\mathrm{d}u}{v(u)}$. The new coordinate system $(\tilde{x}, \tau)$ is isothermal,

$$ds^2 = e^{2\sigma(x)}\left(d\tilde{x}^2 + d\tau^2\right) \quad \text{with} \quad e^{\sigma(x)} = v(x). \tag{52}$$

As a consequence, correlation functions can be written directly with the formalism of Sec. 2, by working in stretched coordinates $\mathrm{x} = (\tilde{x}, \tau)$. To get expressions of correlators in the physical coordinates $(x, \tau)$, one simply has to keep track of Weyl factors: under the Weyl transformation $\mathrm{g} \to e^{2\sigma}\mathrm{g}$, a local operator $\phi(\mathrm{x})$ with scaling dimension $\Delta$ transforms as $\phi(\mathrm{x}) \to e^{-\sigma\Delta}\phi(\mathrm{x})$. For instance, the two-point function of $\phi$ could first be calculated in the coordinate system $(\tilde{x}, \tau)$ using the formalism of Sec. 2, and then be rewritten as

$$\left\langle \phi(x, \tau)\phi(x', \tau') \right\rangle = v(x)^{-\Delta} v(x')^{-\Delta} \left\langle \phi(\tilde{x}, \tau)\phi(\tilde{x}', \tau') \right\rangle. \tag{53}$$

## 3.3 Expansion of the density operator

To relate correlation functions of a microscopic observable $\hat{O}(x)$ to the ones in the IGFF, we need to find an expansion of the form (1) for $\hat{O}(x)$ in terms of local operators in the field theory. This is what we do now, for the local density $\hat{\rho}(x)$. As in Sec. 2, we view the operators as evolving in imaginary time $\tau$. So they are functions of the coordinate $\mathrm{x} = (\tilde{x}, \tau)$, and we will take $\tau = 0$ at the end of the calculation, to get equal-time ground state correlations.

The local operators in the IGFF are the derivative of the height field $\partial_x h$ and the mixed electric-magnetic operators $\mathcal{V}_{\alpha,\beta}(x)$. Operators with non-zero magnetic charge $\beta \neq 0$ cannot appear in the expansion of the local density $\hat{\rho}(x, \tau)$, because they correspond to creation/annihilation processes at point $(x, \tau)$; those will be discussed in more details in Sec. 3.5 below.

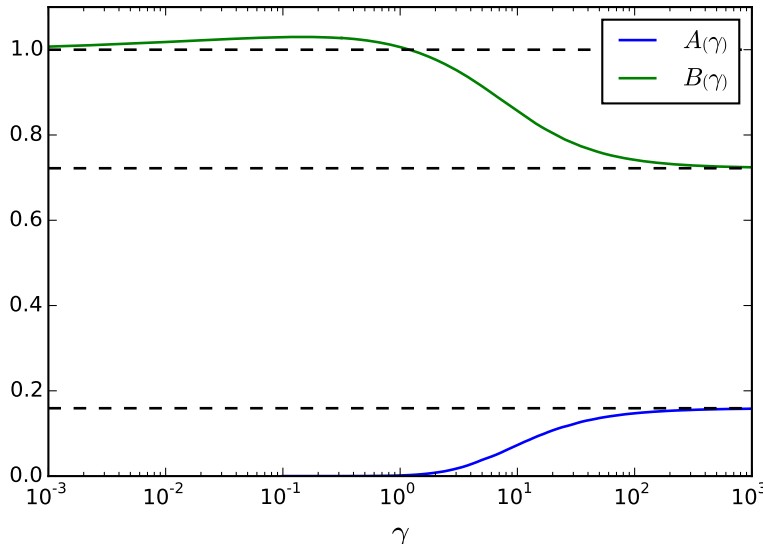

Figure 4: The functions $A(\gamma)$ and $B(\gamma)$ appearing in the expansion of the local density $\hat{\rho}(x)$ and the particle creation operator $\hat{\Psi}^{\dagger}(x)$ in terms of the fields in the IGFF, calculated from Bethe Ansatz form factors [50–52]. See e.g. Refs. [46–49, 58] for details on how to extract such coefficients from form factors. The dashed lines correspond to known asymptotics: $A(\gamma) \to 1/(2\pi)$ when $\gamma \gg 1$, $B(\gamma) \to 1$ when $\gamma \ll 1$, $B(\gamma) \to G^2(3/2)/(2\pi)^{\frac{1}{4}} \simeq 0.722$ when $\gamma \gg 1$ (where $G(.)$ is Barnes' G-function, see Ref. [59]).

So, the local density must have an expansion of the form

$$\hat{\rho}(x,\tau) = \rho_{\mathrm{LDA}}(x) + \frac{1}{2\pi}\partial_x h(x,\tau) + \sum_{p \neq 0} C_{p,0}^{(\hat{\rho})} \mathcal{V}_{p,0}(x,\tau) + \text{descendents}, \tag{54}$$

where the $C_{p,0}^{(\hat{\rho})}$ are dimensionful coefficients that we need to determine, and the "descendents" terms correspond to derivatives of the local operators, which are less local and generate subleading corrections to correlation functions. For simplicity, in this paper we will discard them and keep only the terms $p = \pm 1$ in the sum:

$$\hat{\rho}(x,\tau) = \rho_{\mathrm{LDA}}(x) + \frac{1}{2\pi}\partial_x h(x,\tau) + C_{1,0}^{(\hat{\rho})}\mathcal{V}_{1,0}(x,\tau) + C_{-1,0}^{(\hat{\rho})}\mathcal{V}_{-1,0}(x,\tau)$$
$$+ \text{less relevant terms}. \tag{55}$$

Our task is now to identify the dimensionful coefficients $C_{1,0}^{(\hat{\rho})}$ and $C_{-1,0}^{(\hat{\rho})}$. Once again, we rely on separation of scales, and on the existence of mesoscopic fluid cells in which the system is locally identical to an homogeneous Lieb-Liniger gas. The scaling dimension of the operator $\mathcal{V}_{\pm 1,0}$ is $K$, so, by dimensional analysis, $\left|C_{\pm 1,0}^{(\hat{\rho})}\right| = \langle\hat{\rho}\rangle^{1-K} A(\gamma)$, where $\langle\hat{\rho}\rangle$ is the particle density, and $A(\gamma)$ is a real positive function of the dimensionless interaction parameter $\gamma$. It is also known (see e.g. Ref. [5]) that the phase of the coefficient $C_{\pm 1,0}^{(\hat{\rho})}$ is $e^{\pm i 2k_F x}$ where $k_F = \pi\langle\hat{\rho}\rangle$ is the Fermi momentum. The function $A(\gamma)$ can be calculated from Bethe Ansatz, and is plotted in Fig. 4 (see App. B for details). Since the coefficients $C_{\pm 1,0}^{(\hat{\rho})}$ should depend only on the local properties of the gas, the expression found in the homogeneous case must remain valid also in the inhomogeneous case, replacing $\langle\hat{\rho}\rangle$ and $\gamma$ by $\rho_{\mathrm{LDA}}(x)$ and $\gamma(x)$. We then arrive at the

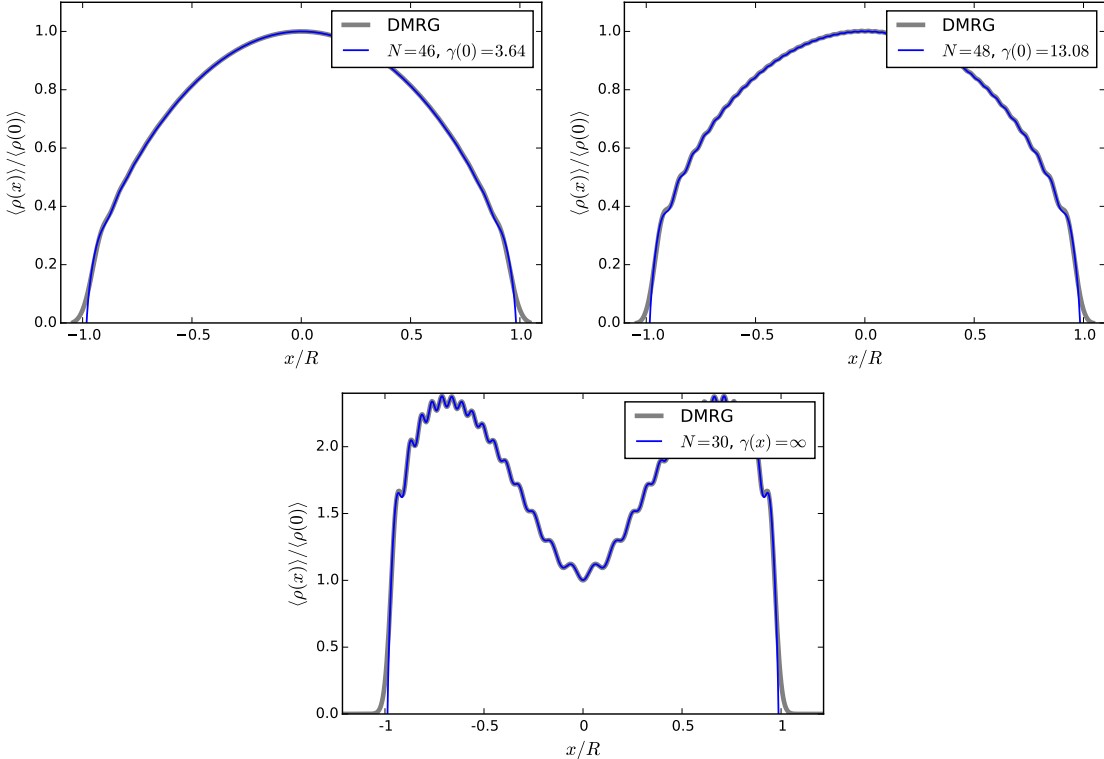

Figure 5: Top row: comparison of the density profiles obtained from formula (58) against the DMRG results for the Lieb-Liniger gas in a harmonic trap (same data as in Fig. 2). We see that the density profile obtained by including the first IGFF correction is in excellent agreement with the exact numerical profile, and that the Friedel oscillations are correctly reproduced at the edge of the trap. Bottom row: to show that the excellent agreement is not restricted to the case of harmonic potentials, we also display the density profile for the Tonks-Girardeau gas (i.e. $\gamma \to +\infty$) in a double-well potential.

expansion of the density operator

$$
\begin{aligned}
\hat{\rho}(x,\tau) \;=\; & \rho_{\mathrm{LDA}}(x) + \frac{1}{2\pi}\partial_x h(x,\tau) + e^{i2\vartheta(x)}\rho_{\mathrm{LDA}}(x)^{1-K(x)}A(x)\mathcal{V}_{1,0}(x,\tau) \\
& + e^{-i2\vartheta(x)}\rho_{\mathrm{LDA}}(x)^{1-K(x)}A(x)\mathcal{V}_{-1,0}(x,\tau).
\end{aligned}
\tag{56}
$$

Here, to lighten the notations, we write $A(x)$ and $K(x)$ instead of $A(\gamma(x))$ and $K(\gamma(x))$. The phase $\vartheta(x)$ is a WKB phase, given by

$$
\vartheta(x) = \pi \int_0^x \rho_{\mathrm{LDA}}(u)\mathrm{d}u - \frac{\pi}{2}.
\tag{57}
$$

It is obtained by requiring that $\partial_x \vartheta(x)$ equals the local Fermi momentum $k_{\mathrm{F}}(x) = \pi\rho_{\mathrm{LDA}}(x)$; the additive constant $\frac{\pi}{2}$ is fixed by an exact calculation in the free fermion case (i.e. the Tonks-Girardeau limit $\gamma \to \infty$), see App. A.

### 3.4 Density profile, and density-density correlation

We now have all the ingredients that are necessary to calculate correlation functions of the local density $\hat{\rho}(x)$. Taking the expectation value of the r.h.s. in Eq. (56), and using the results

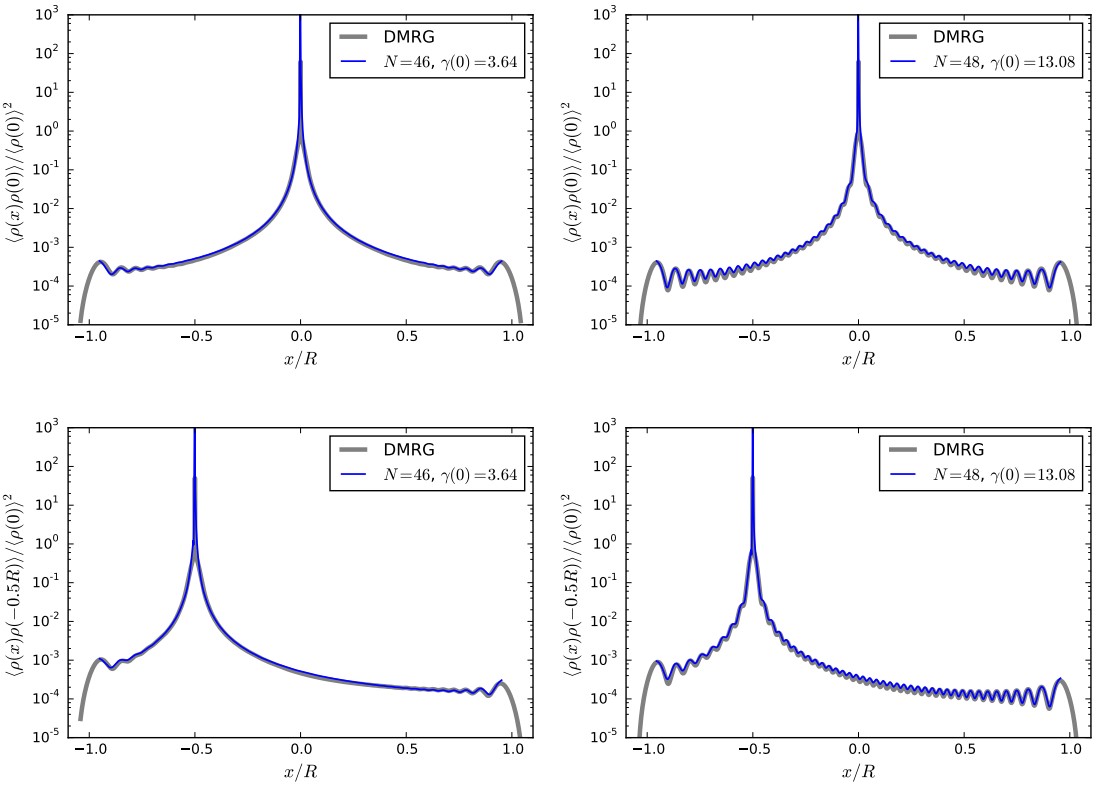

Figure 6: Connected part of density-density correlation function for the Lieb-Liniger gas in an harmonic trap. We compare Eq. (59) to DMRG results (we use the same parameters as in Fig. 2).

of Sec. 2, one finds

$$\langle \hat{\rho}(x) \rangle = \rho_{\text{LDA}}(x) + 2\cos\left[2\vartheta(x)\right] \frac{\rho_{\text{LDA}}(x)^{1-K(x)}}{v(x)^{K(x)}} A(x)\, e^{\frac{1}{2} G_K^{\text{D}}(x)}. \tag{58}$$

This follows from the fact that $\langle \partial_x h \rangle = 0$ and

$$\left\langle \mathcal{V}_{1,0}(x,\tau) \right\rangle = v(x)^{-K(x)} \left\langle \mathcal{V}_{1,0}(\tilde{x},\tau) \right\rangle = v(x)^{-K(x)} e^{\frac{1}{2} G_{[K]}^{\text{D}}(x)},$$

see Eqs. (53) and (42).

In Fig 5, we compare this result to a direct DMRG simulation of the Lieb-Liniger gas. The agreement is excellent. [Another highly non-trivial check for formula (58) is the fact that, in the Tonks-Girardeau limit $\gamma \to +\infty$ and in a harmonic trap, the result is an exact match to the one obtained by evaluating the large-$N$ asymptotics of the Hermite kernel, see App. A for details.] The oscillations of the density are well reproduced by the first subleading corrections from Eq. (55) and are usually interpreted as Friedel oscillations [60, 61].

Next, we use the expansion (56) and the formulae of Sec. 2 to evaluate density-density correlations. [For a study of density-density correlations in the homogeneous case, see e.g.

Ref. [62].] We find, for the connected part,

$$
\begin{aligned}
\left\langle \hat{\rho}(x)\hat{\rho}(x')\right\rangle_{\text{c}} = &-\frac{1}{4\pi^2}\left[v(x)v(x')\right]^{-1}\partial_{\tilde{x}}\partial_{\tilde{x}'}G_K^{\text{D}}(\text{x},\text{x}') \\
&+\frac{1}{\pi}v(x)^{-1}\left[\partial_{\tilde{x}}G_{[K]}^{\text{D}}(\text{x},\text{x}')\right]\sin\left[2\vartheta(x')\right]\frac{\rho_{\text{LDA}}(x')^{1-K(x')}}{v(x')^{K(x')}}A(x')e^{\frac{1}{2}G_{[K]}^{\text{D}}(x')} \\
&+\frac{1}{\pi}v(x')^{-1}\left[\partial_{\tilde{x}'}G_{[K]}^{\text{D}}(\text{x},\text{x}')\right]\sin\left[2\vartheta(x)\right]\frac{\rho_{\text{LDA}}(x)^{1-K(x)}}{v(x)^{K(x)}}A(x)e^{\frac{1}{2}G_{[K]}^{\text{D}}(x)} \\
&+2\left[\left(e^{G_{[K]}^{\text{D}}(\text{x},\text{x}')}-1\right)\cos\left[2\vartheta(x)+2\vartheta(x')\right]+\left(e^{-G_{[K]}^{\text{D}}(\text{x},\text{x}')}-1\right)\cos\left[2\vartheta(x)-2\vartheta(x')\right]\right] \\
&\qquad\times\frac{\rho_{\text{LDA}}(x)^{1-K(x)}}{v(x)^{K(x)}}\frac{\rho_{\text{LDA}}(x')^{1-K(x')}}{v(x')^{K(x')}}A(x)A(x')e^{\frac{1}{2}\left(G_{[K]}^{\text{D}}(x)+G_{[K]}^{\text{D}}(x')\right)},\quad (59)
\end{aligned}
$$

where $\text{x} = (\tilde{x}, \tau)$ and we set $\tau = \tau' = 0$. In Fig. 6, we display a comparison with the density-density correlation obtained from DMRG, as a function of $x$, for two positions $x' = 0$ and $x' = -0.5R$. Again, the agreement is excellent.

### 3.5 The one-particle density matrix

Finally, we will apply the IGFF to the computation of the one-particle density matrix

$$
g_1(x,x') := \left\langle\hat{\Psi}^{\dagger}(x)\hat{\Psi}(x')\right\rangle. \tag{60}
$$

Again, the first step consists in identifying the most relevant field theory operators that appear in the expansion of the creation and annihilation operators $\hat{\Psi}^{\dagger}(x)$ and $\hat{\Psi}(x)$. Here, for simplicity, we restrict ourselves to the leading order, which is given by a single magnetic vertex operator,

$$
\hat{\Psi}(x,\tau) = C_{0,1}^{(\hat{\Psi})}\mathcal{V}_{0,1}(x,\tau) + \text{less relevant operators.} \tag{61}
$$

[Subleading terms will be investigated elsewhere.] The coefficient $C_{0,1}^{(\hat{\Psi})}$ is identified in the same manner as for the density operator: we start by considering the case of homogeneous mesoscopic fluid cells, then go to the inhomogeneous case relying on LDA.

Given that the creation/annihilation operator has dimension $1/2$, and that the magnetic vertex operator has scaling dimension $1/4K$, the amplitude of the coefficient must take the form $\left|C_{0,1}^{(\hat{\Psi})}\right| = \langle\hat{\rho}\rangle^{\frac{2K-1}{4K}}B(\gamma)$ for some function of the dimensionless interaction parameter $B(\gamma)$. This function $B(\gamma)$ is again calculated using form factors formulae, see Refs. [50–52, 63] and Fig. 4. When going to the inhomogeneous case, we know from the homogeneous solution that the coefficient $C_{0,1}^{(\hat{\Psi})}(x)$ does not have a position-dependent phase but can only have a global constant phase, which we can fix to zero, such that $B(\gamma)$ is real and positive. We then have

$$
\begin{cases}
\hat{\Psi}(x,\tau) &= \rho_{\text{LDA}}(x)^{\frac{2K(x)-1}{4K(x)}}B(x)\mathcal{V}_{0,1}(x,\tau), \\
\hat{\Psi}^{\dagger}(x,\tau) &= \rho_{\text{LDA}}(x)^{\frac{2K(x)-1}{4K(x)}}B(x)\mathcal{V}_{0,-1}(x,\tau).
\end{cases} \tag{62}
$$

where we write $B(x)$ instead of $B(\gamma(x))$.

The rest of the calculation is straightforward. We use formula (35), and, taking into account the Weyl factors (53), we obtain

$$
g_1(x,x') = \frac{\rho_{\text{LDA}}(x)^{\frac{2K(x)-1}{4K(x)}}}{v(x)^{\frac{1}{4K(x)}}}\frac{\rho_{\text{LDA}}(x')^{\frac{2K(x')-1}{4K(x')}}}{v(x')^{\frac{1}{4K(x')}}}B(x)B(x')\frac{e^{\frac{1}{2}\left[G_{[1/4K]}^{\text{N}}(\text{x})+G_{[1/4K]}^{\text{N}}(\text{x}')\right]}}{e^{G_{[1/4K]}^{\text{N}}(\text{x},\text{x}')}}, \tag{63}
$$

with $\text{x} = (\tilde{x}, \tau)$, and $\tau = \tau' = 0$. In Fig. 7, we check this formula against DMRG. Even though we only considered the leading order here, we find very good agreement.

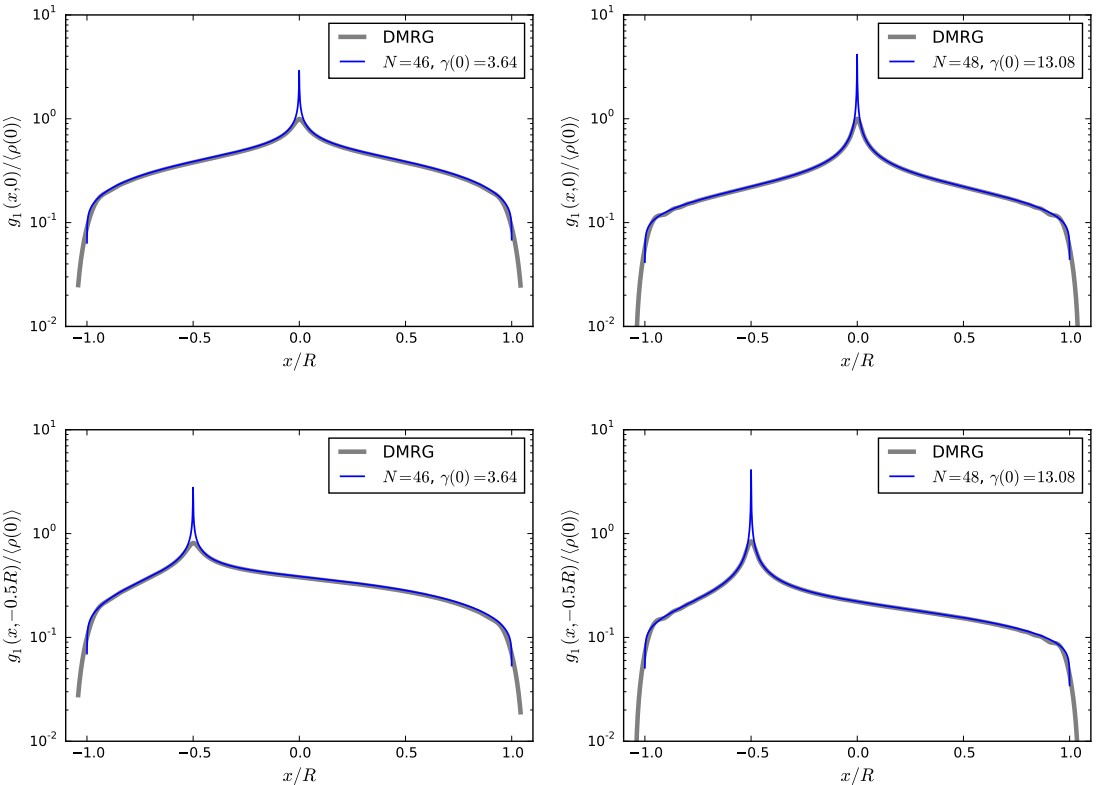

Figure 7: One-particle density matrix for the Lieb-Liniger gas in a harmonic trap, compared to DMRG simulations. (We again use the same parameters as in Fig. 2.)

## 4 Conclusion

The purpose of this paper was to develop the formalism of the IGFF, and provide an exhaustive study of correlations of local observables in that theory. We did so in Sec. 2. Then, in Sec. 3, we explained how, in practice, this formalism gives access to correlation functions of inhomogeneous systems, by focusing on ground state correlations of the Lieb-Liniger gas in a trapping potential.

To conclude this paper, let us mention four directions which, in our opinion, would deserve further investigation.

- in cold atoms experiments, where some correlation functions are measurable [21, 26–28, 32], the gas is at finite temperature. Therefore, it would be very interesting to generalize the results of this paper to finite temperature. Usually, in field theory, working at finite temperature is relatively easy: one simply needs to compactify the imaginary time direction. However, there could be issues related to the boundary of the system: how to properly describe the fluctuations of the particles near the edge of the gas? Those won't be obtained simply by compactifying the time direction in the field theory. It would also be interesting to make the connection with other recent works on trapped 1d quantum gases at finite temperature, for instance Refs. [64–66].

- as mentioned in Sec. 1.1, the inhomogeneous Luttinger liquid also appears in the context of multi-component 1d Fermi gases. Motivated by recent experimental advances [67], it would be interesting to extend the results of Sec. 3 to the case of SU($N$) systems. In this case, the integrable model of interest (the one that replaces the Lieb-Liniger model)

would be the Gaudin-Yang model [45].

- another natural extension of this work would be to tackle time-dependent problems. In the static case studied here, a key role is played by LDA, or hydrostatics, to fix the parameter $K$ and the background metric g in the effective field theory. In a dynamical situation, for instance a breathing Lieb-Liniger gas in a trap [68–71], those parameters would have to be extracted from an hydrodynamic approach. It would be interesting to study how this works in practice, starting with the zero-temperature case. We note that, in a very inspiring paper, Abanov [72] has studied a related problem in imaginary time (see also Ref. [14], on a similar imaginary time problem).

- finally, perhaps the most challenging problem is to understand whether it is possible to have a more general theory of fluctuations and correlations in the recently developed theory of Generalized HydroDynamics (GHD) [73, 74]. So far, the IGFF approach discussed here models only fluctuations of the particle density at zero temperature (and therefore corresponds only to a particular case in the more general GHD framework, dubbed "zero-entropy GHD" in Ref. [75]). In GHD, not only the particle density is expected to fluctuate, but all densities of conserved charges. Perhaps such a theory could take the form of a "fluctuating hydrodynamics" in the spirit of Ref. [76], or perhaps a multi-component version of the IGFF (possibly with arbitrarily large number of components). A step towards correlation functions in GHD has been taken very recently by Doyon in Ref. [77]; it would be a good starting point to understand if/how his results connect to inhomogeneous Luttinger liquids.

## Acknowledgements

We would like to thank P. Calabrese, J.-M. Stéphan and J. Viti for joint work on very closely related topics and for many key discussions on effective field theories of 1d inhomogeneous quantum systems, along with B. Doyon for very useful comments on the manuscript. We are also grateful to V. Alba, D. Bagrets, I. Bouchoule, C. Chatelain, R. Dubessy, F. Essler, B. Estienne, D. Karevski, S. Klevtsov, A. Minguzzi, G. Misguich, J. de Nardis, E. Orignac, V. Pasquier, T. Roscilde, H. Saleur, S. Scopa, S. Sotiriadis, J. Unterberger and T. Yoshimura for stimulating and insightful discussions, and for pointing out relevant references.

YB thanks the Galileo Galilei Institute in Florence, as well as the Institute for Condensed Matter Physics in Lviv for hospitality. JD thanks LPTHE Jussieu (University Paris 6), the ENS Lyon and the ProbabLYon program, Oxford University, Cologne University, the Institut d'Études Scientifiques de Cargèse, and the IPhT Saclay for hospitality.

We acknowledge support from the CNRS-Mission Interdisciplinarité through the Défi IN-FINITI MUSIQ.

The DMRG simulations were performed using the open-source ITensor library [53] .

## A  The Tonks-Girardeau limit

The Tonks-Girardeau (TG) regime is the limit of hard-core repulsion, i.e. $\gamma \to +\infty$. In the special case of a harmonic potential $V(x) = \frac{1}{2}m\omega^2 x^2$, we will show that we recover some known results. But first, let us recall how the exact density can be computed in this case. To keep notations light, we set $\hbar = m = \omega = 1$; then, we have $k_F(x) = v(x) = \pi\rho_{\text{LDA}}(x) = \sqrt{2N - x^2}$.

**Exact density in the harmonic trap —** finding the eigenstates of a single boson confined in the harmonic trap $V(x) = \frac{1}{2}x^2$ is the same problem as solving the quantum harmonic oscillator. The eigenstates take the form

$$\psi_n(x) = \frac{1}{\sqrt{2^n n!}} \pi^{-\frac{1}{4}} e^{-\frac{x^2}{2}} \mathcal{H}_n(x), \tag{64}$$

where the $n^{\text{th.}}$ eigenstate has energy $E_n = \left(n + \frac{1}{2}\right)$ and $\mathcal{H}_n$ is the Hermite polynomial of order $n$. Now, since bosons with infinite-repulsion map to free fermions [78], the groundstate for $N$ bosons can be built by filling up the first $N$ eigenstates. The density is then given by

$$\langle \hat{\rho}(x) \rangle = \sum_{n=0}^{N-1} |\psi_n(x)|^2, \tag{65}$$

which is easily evaluated with the Christoffel-Darboux formula

$$\lim_{x' \to x} \sum_{n=0}^{N-1} \frac{\mathcal{H}_n(x)\mathcal{H}_n(x')}{2^n n!} = \frac{\mathcal{H}'_N(x)\mathcal{H}_{N-1}(x) - \mathcal{H}'_{N-1}(x)\mathcal{H}_N(x)}{2^N (N-1)!}.$$

When $N \gg 1$, this can be put in a more explicit form using the asymptotics of the Hermite polynomials, i.e.

$$e^{-\frac{x^2}{2}} \mathcal{H}_N(x) \sim \frac{2^{\frac{2N+1}{4}} \sqrt{N!}}{(\pi N)^{\frac{1}{4}}} \frac{1}{\sqrt{\sin(\varphi)}} \sin\left(\frac{2N+1}{4}(\sin(2\varphi) - 2\varphi) + \frac{3\pi}{4}\right),$$

where $x = \sqrt{2N+1}\cos(\varphi)$, with $\epsilon \le \varphi \le \pi - \epsilon$ ($\epsilon \to 0$ as $N \to \infty$). Carrying out the asymptotic expansion, we arrive at

$$\langle \hat{\rho}(x) \rangle = \frac{1}{\pi}\sqrt{2N - x^2} - \frac{1}{2\pi} \frac{\cos[2\theta(x)]}{\sqrt{2N}\left(1 - \frac{x^2}{2N}\right)} + \mathcal{O}(1/N), \tag{66}$$

where the phase $\theta(x)$ is the integral of the Fermi momentum $k_F(x)$,

$$\theta(x) = \int_0^x k_F(u)\mathrm{d}u = \frac{x\sqrt{2N - x^2}}{2} - N\arccos\frac{x}{\sqrt{2N}}.$$

After some manipulation, we can also write down an asymptotic expression for the density-density correlation,

$$\begin{aligned}
\langle \hat{\rho}(x)\hat{\rho}(x') \rangle_{\mathrm{c}} &= \frac{1}{2\pi^2} \frac{\left(1 - \frac{xx'}{2N}\right)(x - x')^{-2}}{\sqrt{1 - \frac{x^2}{2N}}\sqrt{1 - \frac{x'^2}{2N}}} \\
&\quad - \frac{1}{2\pi^2} \frac{\left(\sin[2\theta(x)] - \sin[2\theta(x')]\right)(x - x')^{-1}}{\sqrt{2N}\sqrt{1 - \frac{x^2}{2N}}\sqrt{1 - \frac{x'^2}{2N}}} \\
&\quad - \frac{1}{4\pi^2} \frac{\cos[2(\theta(x) + \theta(x'))]\left[\frac{xx'}{2N} - \left(1 + \sqrt{1 - \frac{x^2}{2N}}\sqrt{1 - \frac{x'^2}{2N}}\right)\right]^{-1}}{2N\sqrt{1 - \frac{x^2}{2N}}\sqrt{1 - \frac{x'^2}{2N}}} \\
&\quad + \frac{1}{4\pi^2} \frac{\cos[2(\theta(x) - \theta(x'))]\left[\frac{xx'}{2N} - \left(1 - \sqrt{1 - \frac{x^2}{2N}}\sqrt{1 - \frac{x'^2}{2N}}\right)\right]^{-1}}{2N\sqrt{1 - \frac{x^2}{2N}}\sqrt{1 - \frac{x'^2}{2N}}} + \mathcal{O}(1/N^2). \quad (67)
\end{aligned}$$

Now, when $\gamma \to +\infty$, the Luttinger parameter is constant, $K = 1$ (that is the value for free fermion systems). As a consequence, the action (8) is conformally invariant, and the Green's functions $G_{[1]}^{D}$, $G_{[1/4]}^{N}$ as well as the mixed function $F_{[1,1/4]}^{D,N}$ can be obtained explicitly. Indeed, when the strip is conformally mapped to the upper-half plane, it boils down to an exercise in the method of images,[2] see e.g. [16]. Below, we will show that we recover exactly the results for the average density and for the density-density correlation.

**Density from the (Dirichlet) Green's function $G_{[1]}^{D}$ —** when $\hbar = m = \omega = 1$, the Green's function with Dirichlet boundary conditions takes the explicit form

$$G_{[1]}^{D}(x, x') = \log\left( \frac{\left| \sin\left(\frac{\tilde{x}-\tilde{x}'}{2}\right) \right|^2}{\left| \sin\left(\frac{\tilde{x}+\tilde{x}'}{2}\right) \right|^2} \right), \tag{68}$$

where $x = (\tilde{x}, \tau)$ and $\tau = \tau' = 0$. Its regularization then gives

$$G_{[1]}^{D}(x) = \log\left( \frac{1}{|2\sin(\tilde{x})|^2} \right). \tag{69}$$

The coordinate $\tilde{x}$ is the stretched coordinate from Eq. (51); here, it has an explicit expression, namely $\tilde{x} = \frac{\pi}{2} + \arcsin\frac{x}{2N}$. Finally, with the definition (57), the phase $\vartheta(x)$ reads

$$\vartheta(x) = \theta(x) - \pi.$$

Plugging everything into Eq. (58), we indeed recover the result obtained from the asymptotic expansion of Hermite polynomials (66), with $\lim_{\gamma\to\infty} A(\gamma) = \frac{1}{2\pi}$. The same goes for the density-density correlation (59).

**(Neumann) Green's function $G_{[1/4]}^{N}$ and $g_1(x, x')$ —** similarly, the Green's function with Neumann boundary conditions reads

$$G_{[1/4]}^{N}(x, x') = \log\left( 2\left| \sin\left(\frac{\tilde{x}-\tilde{x}'}{2}\right) \right|^{\frac{1}{2}} \left| \sin\left(\frac{\tilde{x}+\tilde{x}'}{2}\right) \right|^{\frac{1}{2}} \right), \tag{70}$$

and its regularized part takes the form

$$G_{[1/4]}^{N}(x) = \log\left( |2\sin(\tilde{x})|^{\frac{1}{2}} \right), \tag{71}$$

for $x = (\tilde{x}, \tau)$ and $\tau = \tau' = 0$. Plugging these in Eq. (63), we recover the celebrated result for the one-particle density matrix in an harmonic trap, see Refs. [16, 79, 80],

$$g_1(x, x') = B(+\infty)^2 \frac{1}{\sqrt{2\pi}} \frac{|\sin(\tilde{x})|^{\frac{1}{4}} |\sin(\tilde{x}')|^{\frac{1}{4}}}{\left| \sin\left(\frac{\tilde{x}-\tilde{x}'}{2}\right) \right|^{\frac{1}{2}} \left| \sin\left(\frac{\tilde{x}+\tilde{x}'}{2}\right) \right|^{\frac{1}{2}}}, \tag{72}$$

where we know that $\lim_{\gamma\to\infty} B(\gamma)^2 = \frac{G^4(3/2)}{\sqrt{2\pi}}$, with $G(.)$ the Barnes' G-function.

---

[2]The method of images has a wikipedia page: https://en.wikipedia.org/wiki/Method_of_image_charges

**The mixed function** $F_{[1,1/4]}^{\text{D,N}}$ — for completeness, we can also write explicitly the mixed function $F_{[1,1/4]}^{\text{D,N}}(\text{x}, \text{y})$, where now x represents the complex coordinate $\text{x} = \tilde{x} + i\tau$. We find

$$F_{[1,1/4]}^{\text{D,N}}(\text{x}, \text{y}) = \arg\left[ 4\sin\left(\frac{\text{x} - \text{y}}{2}\right) \sin\left(\frac{\text{x} + \bar{\text{y}}}{2}\right) \right], \tag{73}$$

so that its regularized part, taking $\text{x}, \text{y} \to \text{z}$, gives

$$F_{[1,1/4]}^{\text{D,N}}(\text{z}) = \arg\left[ 2\sin(\text{z}) \right]. \tag{74}$$

# B  Extracting the dimensionful coefficients from form factors

In Sec. 3, we have shown how correlation functions can be evaluated using the IGFF. An important ingredient was the set of coefficients $C_j^{\hat{O}}$ that appears in the expansion of a local observable $\hat{O}$ in the microscopic model, in terms of primary operators in the field theory $\phi_j$,

$$\hat{O}(x, \tau) = \sum_j C_j^{(\hat{O})} \phi_j(x, \tau). \tag{75}$$

In this appendix, we explain how the dimensionful coefficients $C_j^{(\hat{O})}$ can be calculated in practice. For similar discussions that have appeared previously in the literature, see e.g. Refs. [46–49, 58].

We work in the homogeneous, translation-invariant, problem, and the field theory is the usual GFF, which is conformally invariant. Thus, we will rely on conformal transformations and on the operator-state correspondence, namely that the operators $\phi_j$ in the CFT correspond to eigenstates of the CFT hamiltonian $\left|\phi_j\right\rangle$.

In fact, Eq. (75) is strictly valid for an infinite system $(x, \tau) \in \mathbb{R}^2$. Since we will rely on numerical evaluation (i.e. we have finite system sizes $L$), our first task is to find a way of taking the limit $L \to \infty$. To do so, we start by making the following assumptions:

- for sufficiently large system sizes $L$, the low-energy excited states of the microscopic hamiltonian $H$ can be unambiguously identified with the ones of the CFT Hamiltonian. In particular, the ground state of $H$ for a system of size $L$, $|0\rangle_L$, is viewed as a microscopic version of the CFT vacuum $|0\rangle$. Similarly, there is a unique eigenstate of $H$, noted $\left|\phi_j\right\rangle_L$, that is viewed as a microscopic version of the CFT state $\left|\phi_j\right\rangle$.

- the form factor in the microscopic model, $_L\left\langle\phi_j\right| \hat{O}(x) |0\rangle_L$, is known for arbitrary $L$.

With this at hand, the dimensionful coefficient $C_j^{(\hat{O})}$ in Eq. (75) is given by

$$C_j^{(\hat{O})} = \lim_{L \to \infty} \left[ \left(\frac{L}{2\pi}\right)^{\Delta_{\phi_j}} \frac{_L\left\langle\phi_j\right| \hat{O}(0) |0\rangle_L}{\sqrt{_L\langle 0 | 0\rangle_L \;\; _L\left\langle\phi_j \middle| \phi_j\right\rangle_L}} \right], \tag{76}$$

where $\Delta_{\phi_j}$ is the scaling dimension of the CFT operator $\phi_j$. This formula is easily obtained as follows.

First, we need to rewrite Eq. (75) for a periodic system $(x, \tau) \in [0, L] \times \mathbb{R}$ with periodic boundary conditions in the $x$-direction. This is done by conformal mapping: the cylinder $x + i\tau \in [0, L] + i\mathbb{R}$ of circumference $L$ is mapped on the infinite plane with the conformal transformation $z = e^{i2\pi\frac{x+i\tau}{L}}$. Then, the r.h.s. in Eq. (75) becomes

$$\sum_j C_j^{(\hat{O})} \left(\frac{2\pi}{L}\right)^{\Delta_{\phi_j}} \phi_j(z, \bar{z}).$$

Next, inserting the l.h.s. of Eq. (75) in $_L\langle\phi_j|\,.\,|0\rangle_L$ and the r.h.s. in $\langle\phi_j|\,.\,|0\rangle$, one gets for $\tau = 0$:

$$\frac{_L\langle\phi_j|\hat{O}(x)|0\rangle_L}{\sqrt{_L\langle 0|0\rangle_L\;\,_L\langle\phi_j|\phi_j\rangle_L}} \simeq C_j^{(\hat{O})}\left(\frac{2\pi}{L}\right)^{\Delta_{\phi_j}}\langle\phi_j|\phi_j(e^{i\frac{2\pi x}{L}})|0\rangle\,,$$

where the denominator in the l.h.s. is the normalization of the two microscopic states and the CFT states and operators are normalized such that $\langle\phi_j|\phi_{j'}(0)|0\rangle = \delta_{j,j'}$ in the plane.

This is an approximation in finite size $L$, but it is expected to become exact in the thermodynamic limit $L \to \infty$, hence the formula (76). [In the case where the operator $\phi_j$ has non-zero spin (or equivalently, if the eigenstate $|\phi_j\rangle_L$ has non-zero momentum), the dimensionful coefficient possesses an $x$-dependent phase, which we dropped from Eq. (76) for simplicity.]

In practice, for the Lieb-Liniger model, we evaluate the coefficients by solving the Bethe equations for a range of particle number $N$, simultaneously varying the length $L = N/\rho$ such that the density $\rho$ is fixed. We solve the Bethe equations numerically,

$$\frac{L}{2\pi}k_j + \frac{1}{2\pi}\sum_{p=1}^{N} i\log\left(\frac{ic + k_j - k_p}{ic - k_j + k_p}\right) = I_j.$$

The eigenstates of the Lieb-Liniger model are indexed by the configurations of Bethe (half-)integers $\{I_1, I_2, \ldots, I_N\}$.

For instance, it is known that the ground state corresponds to the configuration $\{-\frac{N-1}{2}, -\frac{N-3}{2}, \ldots, \frac{N-3}{2}, \frac{N-1}{2}\}$, while the state $|\mathcal{V}_{1,0}\rangle_N$ corresponds to the configuration $\{-\frac{N+1}{2}, -\frac{N-3}{2}, \ldots, \frac{N-3}{2}, \frac{N-1}{2}\}$. More generally, any state in the CFT can be identified with a configuration of Bethe roots close the ground state one, with only a few $I_j$'s that are shifted. See e.g. formula (9.18) in the first chapter of the book by Korepin et al. [57] for more information on the relation between the eigenstates of the LL model and those of the free boson CFT.

Given the ground state and an excited state for a given number of particles $N$, we evaluate the corresponding form factors using the formulae given in Refs. [50–52]. We do this for several system sizes $N$ (or lengths $L = N/\rho$), then perform a polynomial fit in $1/N$ to get a numerical estimate of the limit $N \to \infty$ in formula (76). This is how we obtain the functions $A(\gamma)$ and $B(\gamma)$ displayed in Fig. 4.

In Fig. 8 we display the result from our approach for the one-particle density matrix with the non-universal dimensionful coefficient $B(x)$, against the the result where this coefficient is omitted, $B = 1$. The agreement with the DMRG calculation is much worse in the latter case.

## C   Electrostatics on the 2d lattice

In this appendix, we study classical electrostatics in an inhomogeneous dielectric medium in 2d, in close connection with the discussion of Sec. 2. This is the construction we use to compute the Green's functions $G_{[K]}^{\mathrm{D}}$, $G_{[1/4K]}^{\mathrm{N}}$ and the mixed function $F_{[K,1/4K]}$ numerically; it should therefore help understanding the results of Sec. 2.

Let us start by considering a rectangular lattice whose nodes x are occupied by a discrete height field $h$, and the Luttinger parameter $K$ lives on the edges, see Fig. 9. Equivalently, one can view this as a resistor network, with the field $h$ viewed as an electrostatic potential $V$, and with $K$ viewed as a resistor $R$ on each edge.

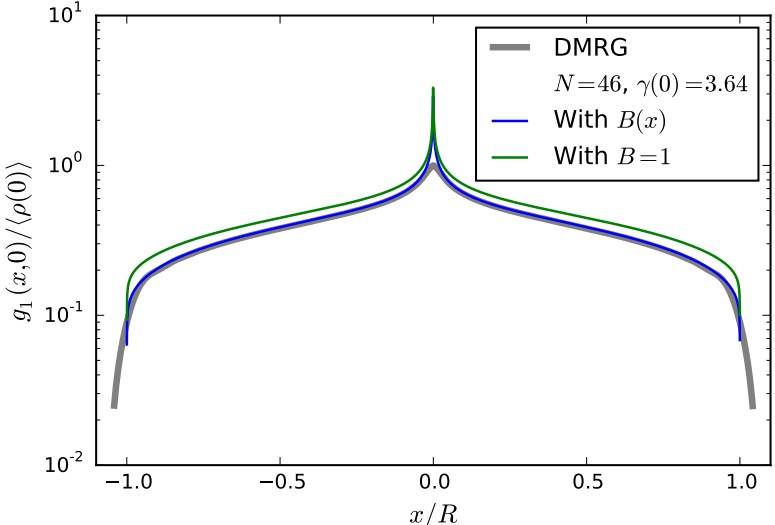

Figure 8: The OPDM from Sec. 3 compared to the result without the non-universal dimensionful coefficient $B(x)$. It shows the importance of such coefficients to quantitatively give the correct results.

**Electric field in an inhomogeneous medium —** as we did in the main text, we can look at the electric field $E$, on an edge $\langle xx' \rangle$ between two neighboring sites x and x',

$$E_{\langle x,x' \rangle} = \frac{h_x - h_{x'}}{|x - x'|},$$

To keep notations light, we restrict to a square lattice with spacing 1. Using Ohm's law, the current on the (oriented) edge $\langle xx' \rangle$ is $I_{\langle xx' \rangle} = \frac{1}{K} E_{\langle x,x' \rangle}$, and the Gauss' law at the vertex x then gives

$$\nabla \cdot \frac{1}{K} \nabla h = \sum_{\substack{x' \text{ neighbor} \\ \text{of x}}} \frac{1}{K_{\langle xx' \rangle}} (h_x - h_{x'}) = 0, \tag{77}$$

in the absence of an electric charge at site $x_i$. If there is an electric charge $\alpha$ on site x, the r.h.s. is proportional to $\alpha$; this is the discrete version of the Gauss law in Eq. (20).

In the absence of a magnetic flux through the plaquettes, the curl of the field $E$ vanishes. For instance, for the sites $x_i$, $x_j$, $x_o$, $x_k$ drawn in Fig. 9,

$$(h_{x_i} - h_{x_j}) + (h_{x_j} - h_{x_o}) + (h_{x_o} - h_{x_k}) + (h_{x_k} - h_{x_i}) = 0,$$

which is the discrete version of Faraday's law in Eq. (20),

$$\nabla \times E = 0. \tag{78}$$

Finally, Dirichlet boundary conditions read $h_x = 0$ for $x \in \partial\Omega$; in terms of the electric field, this implies that the component tangential to the boundary vanishes on $\partial\Omega$,

$$E_\parallel = 0. \tag{79}$$

In electrostatics, this corresponds to the domain $\Omega$ being surrounded by a perfect conductor [55].

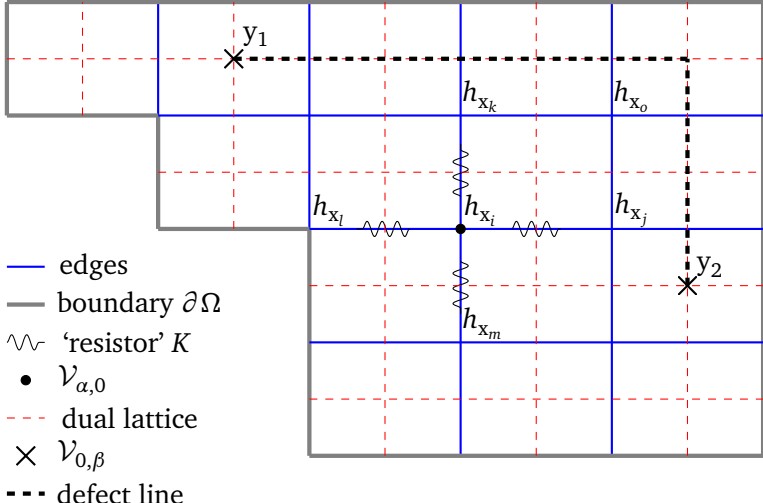

Figure 9: Classical electrostatics in a discretized 2d inhomogeneous medium.

**Magnetic fluxes, electric-magnetic duality —** now, imagine that two plaquettes are pierced by two infinitely thin, constantly increasing, magnetic fluxes in their center, at positions $y_1$ and $y_2$. The fluxes are topological defects around which the field $h$ winds by a constant $\pm 2\pi\beta$. In Fig. 9, this is represented by a defect line linking two defects. When $h$ crosses the defect line, it jumps by $2\pi\beta$. Notice that we have inserted two defects (the two ends of the defect line) with opposite 'magnetic charge' $\pm\beta$, in order to be compatible with the Dirichlet boundary conditions.

In the absence of electric charges on the lattice sites, the electric field now satisfies

$$\begin{cases} \left(\nabla \cdot \dfrac{1}{K} E\right)_x &= 0 \qquad \text{on each site x} \\ (\nabla \times E)_y &= 2\pi\beta_y \qquad \text{on each plaquette y}, \end{cases} \tag{80}$$

where $\beta_y$ is the 'magnetic charge' through each plaquette, here equal to $+\beta$ if $y = y_1$, $-\beta$ if $y = y_2$, and 0 otherwise.

On the 2d lattice, the electric-magnetic duality is easily constructed as follows. The dual field $\tilde{E}$ is defined by a $\pi/2$-rotation of $E$, and a rescaling by $1/(2K)$,

$$\begin{pmatrix} \tilde{E}_1 \\ \tilde{E}_2 \end{pmatrix} = \frac{1}{2K} \begin{pmatrix} E_1 \\ -E_2 \end{pmatrix}, \tag{81}$$

where $E_1$ and $E_2$ are the two components of $E$. This dual field lives on the edges of the lattice, as the original electric field $E$. But one can view $\tilde{E}$ as the discrete gradient of a dual height field $\tilde{h}$, which lives on the vertices of the dual lattice (i.e. the plaquettes of the original lattice), see Fig. 9. Then, the Gauss law reads, for the dual field $\tilde{E}$, $\nabla \cdot 4K\nabla\tilde{h} = \nabla \cdot 4K\tilde{E} = 4\pi\beta_y$, on a plaquette y with magnetic flux $\beta_y$.

Since $\tilde{E}_\perp \propto E_\parallel$, it is also clear that the dual field $\tilde{E}$ satisfies Neumann boundary conditions if $E$ satisfies Dirichlet boundary conditions ($E_\parallel = 0$).

**Mixed electric-magnetic potential $F_{x,y}$ —** finally, we discuss the mixed function $F_{x,y}$ on the lattice. It is defined as the potential felt by an electric charge at site x in the presence of a single magnetic monopole at site y on the dual lattice. More precisely, we start by fixing y, and a defect line that goes from y to an edge at the boundary, see Fig. 10. The height function

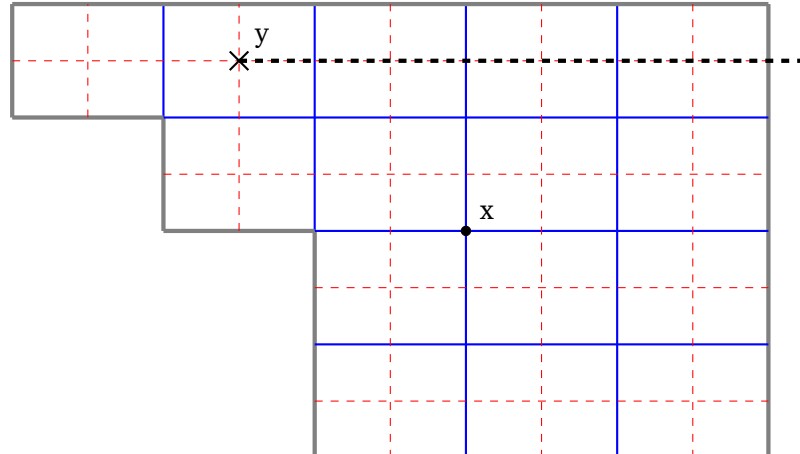

Figure 10: Configuration used for the definition of the mixed function $F_{x,y}$ on the lattice. $F_{x,y}$ is defined as the electric potential felt on vertex x, knowing that the plaquette y is pierced by a flux: $F_{x,y}$ jumps by $\pm 2\pi$ when x crosses the defect line that starts at y (dashed black).

$h_x$ that lives on the vertices has a $2\pi$-discontinuity along the defect line. This means that the discrete gradient of $h$ along an edge $\langle xx' \rangle$, which is usually defined as $(\nabla h)_{\langle xx' \rangle} = h_x - h_{x'}$, is replaced by $(\nabla h)^{(d)}_{\langle xx' \rangle} = \pm 2\pi + h_x - h_{x'}$ on all edges that cross the defect line. The $\pm$ sign is fixed by the orientation of the edge with respect to the defect line. One also fixes a function $f_x$ that lives on the vertices along the boundary $\partial \Omega$, that has a $2\pi$-discontinuity at the edge where the defect line crosses the boundary, see Fig. 10. The mixed function $F_{x,y}$ is then defined as the height function $h_x$ that has the right discontinuity along the defect line, and satisfies the Dirichlet boundary conditions $h_x = f_x$ along the boundary.

In other words, the mixed function $F_{x,y}$ is defined as the solution of the linear problem

$$\begin{cases} \nabla_x \cdot \dfrac{1}{K} \nabla_x F_{x_i, y_i} &= 0, \\ F_{x,y} &= f_x \quad \text{if} \quad x \in \partial\Omega, \end{cases} \tag{82}$$

where the definition of $\nabla F$ is replaced by $(\nabla F)^{(d)}$ on edges crossed by the defect line. This is the lattice version of Eq. (38) in the main text.

So far, we have regarded $F_{x,y}$ as a function of x, defined for some fixed y. But it is interesting to see that it also satisfies a set of dual constraints, as a function of the variable y,

$$\begin{cases} \nabla_y \cdot K \nabla_y F_{x,y} &= 0, \\ (\nabla_y F_{x,y})_\perp &= (\nabla f_y)_\parallel \quad \text{if} \quad y \in \partial\Omega, \end{cases} \tag{83}$$

which are the discrete version of Eq. (40). We now show that the first equation in (83) follows from (82); we leave the second one (the boundary condition) as an exercise to the reader.

First, we note that, for two neighboring plaquettes y and y$'$, the discrete gradient $F_{x,y} - F_{x,y'}$ is the electrostatic potential created by a short defect line on the dual edge $\langle yy' \rangle$, viewed at point x. This is illustrated in the following picture,

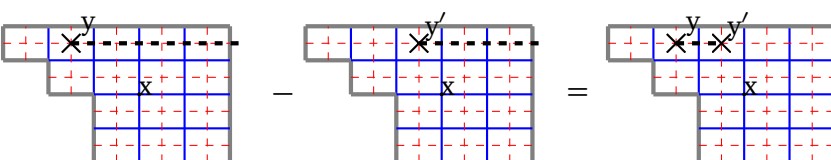

Thus, the combination $Q_{\mathrm{x}} := \nabla_{\mathrm{y}} \cdot K \nabla_{\mathrm{y}} F_{\mathrm{x,y}}$ corresponds to a sum of four terms, which we can view as the potential created by four short defect lines around y:

$$Q_{\mathrm{x}} := \nabla_{\mathrm{y}} \cdot K \nabla_{\mathrm{y}} F_{\mathrm{x,y}} = \quad$$ 

Now comes the crucial observation: this combination $Q_{\mathrm{x}}$ satisfies $\nabla_{\mathrm{x}} \cdot \frac{1}{K} \nabla_{\mathrm{x}} Q = 0$ *for all sites* $\mathrm{x} \in \Omega$. For sites that are sufficiently far from the plaquette y, this is obvious, and it simply follows from the fact that $F_{\mathrm{x,y}}$ satisfies this equation. However, when x is one of the four corners of the plaquette y, one must be careful with the $\pm 2\pi$ discontinuities. Writing the four terms appearing in the explicit expression of the discrete operator $\nabla_{\mathrm{x}} \cdot \frac{1}{K} \nabla_{\mathrm{x}}$, one sees that exactly two of them correspond to terms on edges that cross the defects:

$$\nabla_{\mathrm{x}} \cdot \frac{1}{K} \nabla_{\mathrm{x}} Q_{\mathrm{x}} = \quad = 0.$$

The $\pm 2\pi$ jumps coming from those two crossings cancel, and the relation $\nabla_{\mathrm{x}} \cdot \frac{1}{K} \nabla_{\mathrm{x}} Q = 0$ holds, as claimed.

In addition, it is clear that $Q_{\mathrm{x}} = 0$ along the boundary $\mathrm{x} \in \partial \Omega$. Those two facts imply that $Q_{\mathrm{x}}$ is identically zero, so $\nabla_{\mathrm{y}} \cdot K \nabla_{\mathrm{y}} F_{\mathrm{x,y}} = 0$ as claimed in (83).

# D   DMRG setup

In this work, Density Matrix Renormalization Group (DMRG) simulation was performed using the open-source C++ library ITensor [53]. The Lieb-Liniger model can be discretized in terms of the XXZ Heisenberg spin chain in its low-density regime [81], and in DMRG, this is the most usual way to simulate the LL model (along with the Bose-Hubbard model) [82–85]. Under this mapping, the XXZ Hamiltonian reads

$$H_{\mathrm{XXZ}} = -\frac{J}{2} \sum_{j=1}^{n-1} \sigma_j^+ \sigma_{j+1}^- + \sigma_j^- \sigma_{j+1}^+ + \sum_{j=1}^{n} (J - \mu + V(ja_0)) \sigma_j^z - \sum_{j=1}^{n-1} \frac{J}{1 + U/2J} \sigma_j^z \sigma_{j+1}^z, \quad (84)$$

where $j$ labels the sites, $a_0$ is the lattice spacing, $n$ is the total number of sites, $J = \hbar^2/ma_0^2$ and $U = g/a_0$. In the low-density regime $a_0 \ll \rho_{\mathrm{max}}^{-1}$, the continuous position corresponds to $ja_0 \to x$.

We denote by $|\phi\rangle$ the ground state of Hamiltonian (84). The correlation functions of the LL model are then easily computed in terms of the Pauli matrices $\sigma_j$. The connected part of the density-density correlation is given by

$$\langle \hat{\rho}(x) \hat{\rho}(x') \rangle_{\mathrm{c}} = \langle \phi | \sigma_j^z \sigma_{j'}^z | \phi \rangle - \langle \phi | \sigma_j^z | \phi \rangle \langle \phi | \sigma_{j'}^z | \phi \rangle. \quad (85)$$

Similarly, the one-particle density matrix can be computed in terms of the raising and lowering operators,

$$g_1(x, x') = \langle \phi | \sigma_j^+ \sigma_{j'}^- | \phi \rangle. \quad (86)$$

In order to check that the low-density regime is correctly fulfilled, we can cook up some criterion. Indeed, performing DMRG in the homogeneous gas, we can extract numerically the

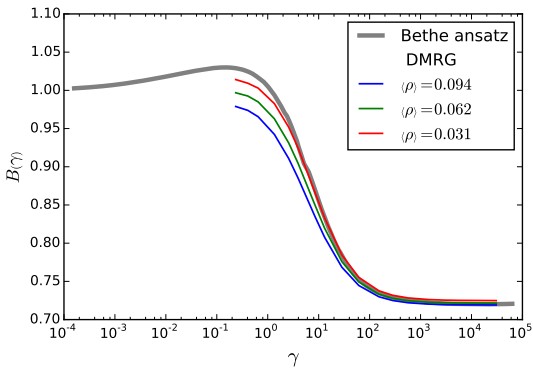

(a) Coefficient $B(\gamma)$ extracted from DMRG simulations compared to the result obtained from algebraic Bethe ansatz form factors.

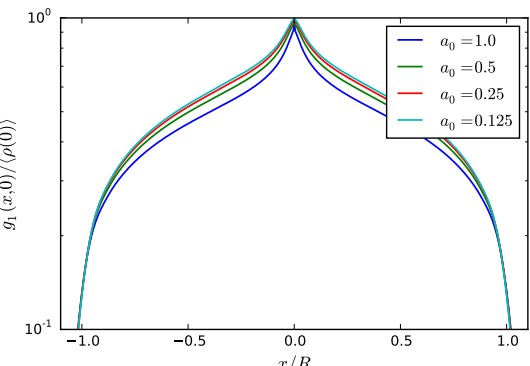

(b) Convergence of the OPDM as the lattice spacing $a_0$ is decreased.

Figure 11: Criterion we use to check that the low-density regime is reached. We see that the discretization must be increased as we want to simulation the LL model with smaller interaction parameter $\gamma$.

prefactors $A(\gamma)$ and $C(\gamma)$ from the simulation, and check that they match with the (exact) ones calculated via algebraic Bethe ansatz. In Fig. 11a, we see that, as $\gamma$ gets smaller, the two results match as the density gets lower. This seems consistent with the fact that the Bethe ansatz form factors are calculated for $\langle \hat{\rho} \rangle \to 0$. However, since we want to simulate systems with large numbers of particles, we can just as well increase the discretization. Concretely, we set the lattice spacing to $a_0 = 1$ for $n = 512$ sites; the, results seem to converge for $a_0$ decreased by at least one order of magnitude, see Fig. 11b. In Sec. 3, simulations were performed on a lattice of $n = 4096$ sites.

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
