# Peer review of "The Inhomogeneous Gaussian Free Field, with application to ground state correlations of trapped 1d Bose gases"

_SciPost Physics, doi:SciPost Phys. 4, 037 (2018)_

## Round 1 · Referee Report · Anonymous · 2018-2-2

Strengths

- The paper presents solid and high-quality results
- It is interesting and very well written
- It constitutes a good analysis of the Bose gas inside a trap, which is a long-standing problem in cold atoms physics

Weaknesses

- Some clarifications needed - see report
- The physical aspects of the results could be discussed in more depth - see report
- Minor, formal points - see report

Report

1- The authors introduce an algorithm to go from the density of the gas given by the LDA approximation, to then obtain the parameters K(x),g(x) and then finally the expectation value of the density operator <rho(x)>. If I understand correctly the algorithms reads as (and I believe it would be good to show such a diagram in the manuscript)
\rho_{LDA}(x) -> \gamma(x) -> K(x),g(x) -> < \rho(x) >
If this is correct then I wonder if it should be intended as a recursive algorithm, namely once one has found an expression for < \rho(x) >, this can be used as a new input of the algorithm (more correct than \rho_{LDA}(x)).

2- Formula 3.12: what is the reason to stop to the first terms of the expansion? Including the other terms is much more complicated?

3- These terms (previous point) are usually related to umklapp excitations in the microscopic model. How this association generalizes in the inhomogeneous case? Is there any physical picture that allows seeing the terms in eq 3.16 as umklapp excitations between the point x and the point 0 of the system?

4- The authors seem to focus only on harmonic trapping potential. Including non-harmonic terms does represent a problem for their calculation or not? It would be also interesting to see how these change the correlation function.

5- In Fig. 5 it seems that the IGFF is in good agreement with DMRG. However, at the two edges, large deviations between the two results appear. Maybe the authors can comment on this effect.

6- Is there any physical interpretation for the oscillations on the density profile that seem to increase for large values of gamma(0)?

7- The authors mention in the introduction that the results of refs [38,38,39] are incorrect, while their result is instead correct. It would be really good to have at least one plot where the two results are shown together with the DMRG simulation, in order to check how much the approximation done in those references is far from being true.

8- in Appendix B it would be useful to have some references where the correspondence between CFT states and Bethe Ansatz excited states is described (in particular for the correct choice of quantum numbers).

9- In this regard, together with references 45,46, the authors should also refer to the works of the french group on prefactors in integrable models, see for example N. Kitanine, K. K. Kozlowski, J. M. Maillet, N. A. Slavnov, V. Terras, J. Stat. Mech. (2011) P12010 and N. Kitanine, K. K. Kozlowski, J. M. Maillet, N. A. Slavnov, V. Terras, J. Stat. Mech., P09001 (2012).

Requested changes

see report

  • validity: top
  • significance: high
  • originality: high
  • clarity: high
  • formatting: excellent
  • grammar: excellent

Author:  Yannis Brun  on 2018-05-17  [id 254]

(in reply to Report 1 on 2018-02-02)
Category:
answer to question

We are grateful to the referee for their positive feedback on the manuscript. Below we address the remarks and issues raised in the report.

"1- The authors introduce an algorithm to go from the density of the gas given by the LDA approximation, to then obtain the parameters $K(x)$, $g(x)$ and then finally the expectation value of the density operator $\left< \rho(x) \right>$. If I understand correctly the algorithms reads as (and I believe it would be good to show such a diagram in the manuscript) $\rho_{LDA}(x) \to \gamma(x) \to K(x),~g(x) \to \left< \rho(x) \right>$. If this is correct then I wonder if it should be intended as a recursive algorithm, namely once one has found an expression for $\left< \rho(x) \right>$, this can be used as a new input of the algorithm (more correct than $\rho_{LDA}(x)$)."

$\rightarrow$ Indeed, the density profile given by the LDA is used to extract interaction and the Luttinger parameters, $\gamma(x)$, $K(x)$ and $v(x)$. However, our primary goal is not to get an ‘improved’ density profile $\left< \rho(x) \right>$, it is, more generally, to be able to calculate long-distance correlation functions in a field theory setup. It turns out that the one-point function of the density operator is one of these correlation functions, and that the approach we develop naturally produces corrections to the LDA profile. In particular, it reproduces the Friedel oscillations near the edges of the trap, that are absent from ’bare’ LDA.

The approach is not intended to be a recursive algorithm. One reason for this is that the approach is based on the assumption that local parameters such as $gamma(x)$, $K(x)$, $v(x)$ vary slowly at the microscopic scale (the inter-particle distance). This is true for parameters extracted from LDA, because LDA is based on separation of scales, but it would no longer be true after the corrections are included, because those corrections include terms that are no longer slowly varying at the microscopic scale (e.g. the Friedel oscillations).

"2- Formula (3.12): what is the reason to stop to the first terms of the expansion? Including the other terms is much more complicated?"

$\rightarrow$ Yes. Although conceptually higher-order terms are similar, in practice, including the latter becomes more tedious. One has to extract other dimensionful coefficients from Bethe Ansatz form factors, and expressions for correlation functions include much more terms (see Eq. (3.16), which is already quite long, while being based only on the first terms in the expansion). Another motivation for stopping at the terms with $p = \pm 1$ is that these are sufficient to match exactly the asymptotic formulae derived in Appendix A for the Tonks-Girardeau gas in a harmonic trap using known asymptotic formulae for Hermite polynomials, see Eqs. (A.3) and (A.4).

"3- These terms (previous point) are usually related to umklapp excitations in the microscopic model. How this association generalizes in the inhomogeneous case? Is there any physical picture that allows seeing the terms in Eq. (3.16) as umklapp excitations between the point x and the point 0 of the system?"

$\rightarrow$ It seems to us that these cannot be related to Umklapp excitations, which are linked to the presence of a lattice and yields oscillatory terms with wavelength $4k_F$ (see for instance Giarmachi's book [4], section 4.2). Here, the terms with $p = \pm 1$ give oscillations with $2k_F$ and can be interpreted as Friedel oscillations.

"4- The authors seem to focus only on harmonic trapping potential. Including non-harmonic terms does represent a problem for their calculation or not? It would be also interesting to see how these change the correlation function."

$\rightarrow$ In fact, the approach allows to consider an arbitrary trapping potential and we show in Fig. 5 that it works for the density profile of the Tonks-Girardeau gas in a double-well potential. The shape of the potential is simply taken into account in the stretched coordinate $\tilde{x}$ (see Eq. (3.8)), which has to be evaluated numerically in general. Thus, one can compute correlation functions for any shape of the trap. We chose to put the harmonic trap in the spotlight as it seems to be the most relevant to experimental realizations.

"5- In Fig. 5 it seems that the IGFF is in good agreement with DMRG. However, at the two edges, large deviations between the two results appear. Maybe the authors can comment on this effect."

$\rightarrow$ This is easily understood as follows. As in any field theoretic approach, we are trying to describe large-scale correlations, at distances much larger than the interparticle distance. But, near the edges of the cloud, the assumptions that underlie this approach break down. Thus, the fact that the agreement between a field theory calculation and the true microscopic simulation is getting worse near the edges is not surprising.

In fact, for us, it is the other way around: we are surprised that the field theory approach seems to work so close to the edge (i.e. even near the leftmost or rightmost individual particle in the trap). We do not understand why it works so well.

"6- Is there any physical interpretation for the oscillations on the density profile that seem to increase for large values of $\gamma(0)$?"

$\rightarrow$ In the literature, these oscillations are usually interpreted as Friedel oscillations (see for instance [R. Egger and H. Grabert, PRL 75, 3505, 1995] or [M. Cazalilla, EPL 59 (6), 793, 2002]). For the trapped Lieb-Liniger model at finite interaction strength, they appear because the system has a Fermi surface. When the repulsion between the bosons gets larger, the bosons exhibit more fermionic character and the oscillations becomes larger. Conversely, when the interaction goes to zero, the Fermi surface disappears and the oscillations vanish. This is a ‘physical’ interpretation. Now, at a more formal level, there is a combination of two facts. First, in the Luttinger liquid, the amplitude of the oscillations decay (as a function of the distance to the edge) as a power-law, with an exponent that increases for larger $K$. This effect is well-known in the literature, see for instance the Refs. by Egger-Grabert and Cazalilla above. Second —and this is a new observation, as far as we are aware—, the oscillations are also killed for small gamma by the coefficient $A(\gamma)$ that quickly goes to zero for small interaction values.

7- The authors mention in the introduction that the results of refs [36,38,39] are incorrect, while their result is instead correct. It would be really good to have at least one plot where the two results are shown together with the DMRG simulation, in order to check how much the approximation done in those references is far from being true.

$\rightarrow$ We now provide in Appendix B a plot for the OPDM, showing the result from our approach with the non-universal dimensionful coefficient $B(x)$, compared to the result where those coefficients are omitted. Without those coefficients, the agreement with the DMRG calculation is much worse.

"8- In Appendix B it would be useful to have some references where the correspondence between CFT states and Bethe Ansatz excited states is described (in particular for the correct choice of quantum numbers)."

$\rightarrow$ Although this is certainly well-known among experts, we confess that we do not know a reference where the correspondence between CFT states and Bethe Ansatz states is discussed in details. One reference where this is sketched is the book of Korepin, Bogoliubov and Izergin (ref. [56] in our manuscript): there, the structure of the space of excitations above the ground state for the Lieb-Liniger is discussed in chapter 1, and the correspondence with CFT states is obtained via formula (9.18) (where $v$ and $\mathcal{Z} = \sqrt{K}$ are the two Luttinger parameters). The formula gives the finite-size correction to the energy of the eigenstates in the thermodynamic limit, and is in agreement with the spectrum of the free boson CFT. The quantum numbers (i.e. total momentum and number of particles) appear explicitly in that formula.

We added a sentence referring to formula (9.18) in chapter 1 of Korepin, Bogoliubov and Izergin in our Appendix B.

"9- In this regard, together with references 45,46, the authors should also refer to the works of the french group on prefactors in integrable models, see for example N. Kitanine, K. K. Kozlowski, J. M. Maillet, N. A. Slavnov, V. Terras, J. Stat. Mech. (2011) P12010 and N. Kitanine, K. K. Kozlowski, J. M. Maillet, N. A. Slavnov, V. Terras, J. Stat. Mech., P09001 (2012)."

$\rightarrow$ Thanks for pointing out these references, we have included them.

---

## Round 2 · Author Response

We thank the editor for taking in charge the submission and refereeing of our manuscript. We here resubmit a new version of the manuscript with minor modifications, see answer to report 1.

---

## Editorial Decision

published